# Rethinking Attention in Spiking Transformers: Overcoming Density Bias with Set Similarity

JinGyo Lim [1]    SeungGyu Jeong [1]    Seong-Eun Kim [1]

## Abstract

Recent Spiking Transformer models have explored a variety of attention mechanisms beyond standard dot-product formulations. However, many existing similarity-based spiking attention formulations remain inherently sensitive to firing density, causing neurons with high spike rates to dominate attention scores regardless of semantic relevance. This density bias is particularly problematic in event-driven spiking representations, where sparse spike patterns often carry critical information. To address this limitation, we rethink spiking attention from a set-theoretic perspective. We propose DiceFormer, a novel Spiking Transformer architecture driven by Spike Dice Attention (SDA). Unlike traditional approaches, SDA replaces density-sensitive measures with a set similarity function derived from the Dice coefficient. By explicitly normalizing for firing density, SDA focuses on spike co-occurrence rather than high firing rates. We primarily evaluate DiceFormer on the challenging audio domain, where spike sparsity varies substantially across inputs. On AudioSet-20k, DiceFormer achieves an SNN SOTA mAP of 0.161 with 54.3M parameters, outperforming prior SNN-based approaches and substantially narrowing the performance gap with ANN-based models. We also introduce Lin-SDA, a linearized version for computation efficiency, while achieving performance comparable to SDA. To provide initial evidence of SDA's broader applicability, we additionally evaluate SDA on CIFAR-100 as a preliminary image-domain benchmark.

## 1. Introduction

Artificial Neural Networks (ANNs), especially Transformer architectures, have propelled major advances across computer vision, audio, and speech. In computer vision, Transformer-based models drive progress in image classification (Yuan et al., 2021; Liu et al., 2021), object detection (Carion et al., 2020; Zhu et al., 2020), and semantic segmentation (Wang et al., 2021; Yuan et al., 2022); in audio, they enable sound event classification (Gong et al., 2021; 2022; Huang et al., 2022), source separation (Xu et al., 2025; Subakan et al., 2021); and in speech, large attention-based encoders have become standard for automatic speech recognition and representation learning (Baevski et al., 2020; Hsu et al., 2021; Radford et al., 2023). However, as the scale and complexity of ANN models increase, they face a significant challenge of substantial energy consumption. To address this, Spiking Neural Networks (SNNs) (Maass, 1997; Izhikevich, 2003; Masquelier et al., 2008) have emerged as a promising alternative, offering an event-driven computational paradigm inspired by the human brain. Operating with discrete spikes, SNNs enable much greater energy efficiency, motivating their use in applications ranging from bearing fault diagnosis (Lim & Kim, 2025a) to biosignal processing (Kang et al., 2025). However, despite this promise, SNNs have consistently exhibited lower accuracy compared to ANNs, and recent progress in bridging this gap has been predominantly concentrated in the image domain.

Much of this progress stems from adapting Vision Transformer (ViT) (Dosovitskiy et al., 2021) architectures to the spiking setting. In image classification, models such as Spikformer (Zhou et al., 2023), Spike-driven Transformer (Yao et al., 2023a), and QKFormer (Zhou et al., 2024) have achieved strong performance while retaining energy efficiency. Similarly, Spike-driven Transformer V2 (Yao et al., 2024) extends these gains to dense prediction tasks such as semantic segmentation and object detection. Further successes have been reported in generative modeling, with spiking variational autoencoders (VAEs) (Kamata et al., 2022) and spiking diffusion models (Cao et al., 2024).

Despite these advances, progress has remained largely confined to the visual domain, and principled attention mechanisms for spike signals are still not well established. Exist-

[1]Department of Artificial Intelligence Applications, Seoul National University of Science and Technology, Seoul, Republic of Korea. Correspondence to: Seong-Eun Kim <sekim@seoultech.ac.kr>.

*Proceedings of the 43$^{rd}$ International Conference on Machine Learning*, Seoul, South Korea. PMLR 306, 2026. Copyright 2026 by the author(s).

ing SNN-ViT models (Yao et al., 2023a; 2024; Zhou et al., 2024) often use spike-based scoring that is density-unaware, meaning attention scores are computed without explicitly accounting for the spike density (firing rate) of the query and key vectors. As a result, scores can be biased toward high-density spikes, blurring the distinction between firing activity and true spike similarity and limiting the capture of meaningful spike-based relationships.

Beyond these attention mechanism-level limitations, extending spike-based transformers to the audio domain requires addressing the distinct nature of audio inputs. Although Mel-spectrograms are the standard 2D representation for audio, they fundamentally differ from natural images in their frequency–temporal structure (Ahmed et al., 2024; Luan et al., 2025), thus requiring tailored processing strategies.

To overcome these limitations, we propose **DiceFormer**, to the best of our knowledge, the first hierarchical SNN Transformer designed to effectively capture frequency-temporal dependencies for general audio classification. At its core is **Spike Dice Attention (SDA)**, a linear-time mechanism robust to spike-density variation, extended by Spike Audio Dice Attention (SADA) to capture frequency–temporal structures. We also introduce Lin-SDA, a variant optimized for computational efficiency. Trained from scratch, DiceFormer achieves new SNN SOTA performance on AudioSet (Gemmeke et al., 2017) and ESC-50 (Piczak, 2015), while remaining competitive on Speech Commands V2 (Warden, 2018). While our primary focus is audio, we also include CIFAR-100 as a preliminary vision benchmark to assess the compatibility of SDA with image inputs. Furthermore, DiceFormer maintains competitive performance with ANNs while significantly reducing energy consumption, highlighting its potential as an efficient SNN-based approach for audio classification.

Our main contributions are summarized as follows:

1. **Spike Dice Attention (SDA).** We propose SDA, a linear-complexity $\mathcal{O}(ND)$ mechanism that explicitly normalizes for spike density to yield robust similarity estimates. We also present **Lin-SDA**, a computation-efficient variant.

2. **Theoretical Analysis of Spike Density Impact.** We provide both mathematical analysis and empirical evidence to identify the density bias in existing methods and validate the effectiveness of our density-aware formulation.

3. **Extending SNN Transformers to Audio.** We introduce DiceFormer as an SNN architecture tailored to the audio domain, and show that training from scratch substantially narrows the performance gap to ANN Transformers while preserving energy efficiency.

## 2. Related Work

**Audio Transformers.** Early progress in audio classification was driven by CNN-based architectures such as PANNs (Kong et al., 2020), which provided strong baselines through convolutional feature extraction. However, the field has undergone a paradigm shift toward Transformer-based architectures, which are better suited to capture long-range dependencies in sequential data. A pioneering step in this direction was the Audio Spectrogram Transformer (AST) (Gong et al., 2021), which adapted the Vision Transformer (ViT) paradigm to audio by treating spectrograms as two-dimensional images. AST divides the spectrogram into overlapping patches and processes them as a token sequence with a Transformer encoder, enabling the model to capture global frequency–temporal correlations. Additionally, SSAST (Gong et al., 2022) introduced self-supervised pretraining using Masked Spectrogram Patch Modeling (MSPM), involving masking, reconstructing, and contrasting spectrogram patches. To further align architecture designs with the unique properties of audio data, recent works have focused on refining feature interactions. Notably, DTF-AT (Alex et al., 2024) explicitly decoupled temporal and frequency modeling, creating a dual-branch architecture that advanced performance by learning these two dimensions separately before combining them into a unified representation.

**SNN Vision Transformers.** The adaptation of the Vision Transformer (ViT) (Dosovitskiy et al., 2021) to Spiking Neural Networks (SNNs) has spurred a new generation of high-performance spiking architectures. Spikformer (Zhou et al., 2023) was the first to propose an SNN-ViT, introducing dot-product-based spiking self-attention that demonstrated the feasibility of combining spiking dynamics with Transformer architectures. Building on this, Spike-driven Transformer (Yao et al., 2023a) proposed a Hadamard-product-based attention mechanism, achieving linear computational complexity and significantly narrowing the performance gap between SNNs and ANNs on image classification. The framework was later extended by Spike-driven Transformer V2 (Yao et al., 2024), which introduced a hierarchical architecture combined with dot-product-based attention to achieve strong results in dense prediction tasks, including object detection and semantic segmentation. Most recently, QKFormer (Zhou et al., 2024) advanced this line of research by combining hierarchical representations with a more efficient attention mechanism, achieving SOTA accuracy for SNNs on ImageNet-1K (Deng et al., 2009) and demonstrating performance competitive with ANN baselines (Dosovitskiy et al., 2021; Touvron et al., 2021), yet such effective designs have barely been explored in the audio domain.

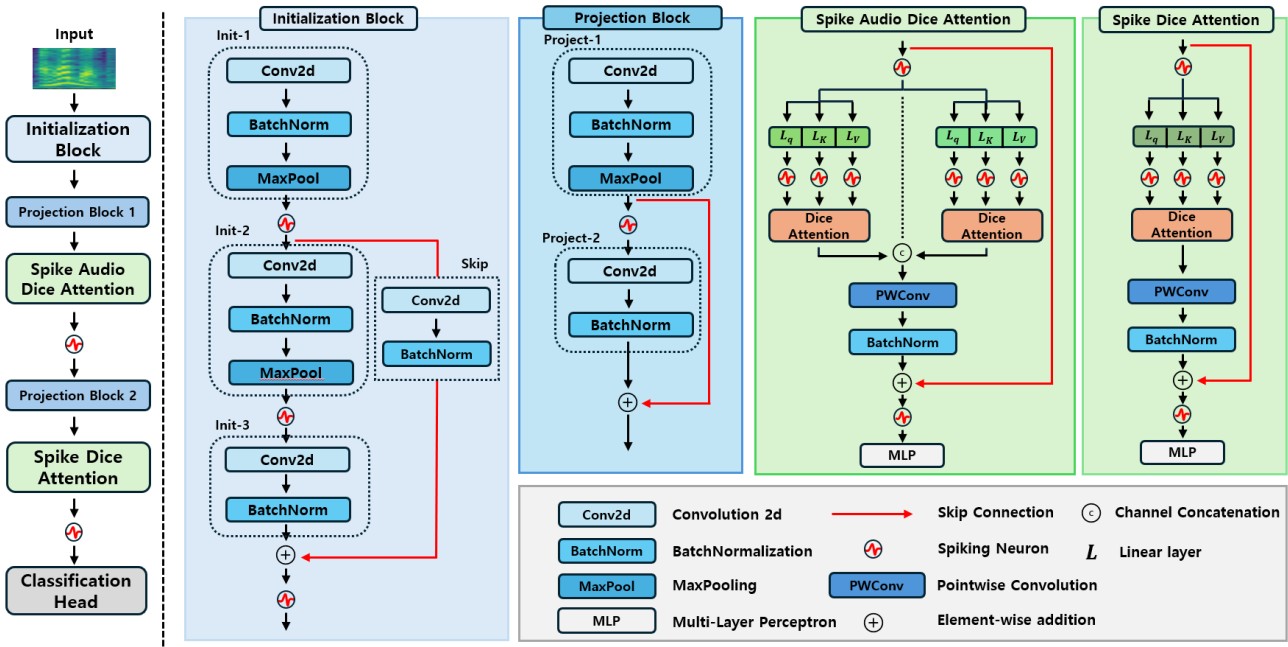

Figure 1. Overall architecture of the proposed DiceFormer. The model processes the input through five main stages: an Initialization Block, Projection Conv Blocks, a SADA module, an SDA module, and a final Classification Head.

## 3. Method

The architecture of our proposed DiceFormer, illustrated in Figure 1, is a hierarchical network designed to learn features at multiple scales. As a Spiking Neural Network, DiceFormer integrates Spiking Neurons (SN) after its main computational layers to convert continuous-valued features into binary spikes for event-driven processing.

**Spiking Neuron** To convert continuous-valued features into binary spikes, we employ the Parametric Leaky Integrate-and-Fire (PLIF) neuron model (Fang et al., 2021). Its dynamics at each time-step $t$ are defined as

$$H[t] = V[t-1] + \frac{1}{\tau}\left(X[t] - (V[t-1] - V_{\text{reset}})\right) \quad (1)$$

$$S[t] = \text{Heav}\big(H[t] - V_{\text{th}}\big) \quad (2)$$

$$V[t] = H[t]\,(1 - S[t]) + S[t]\,V_{\text{reset}}, \quad (3)$$

where $X[t]$ is the input current, $V_{\text{th}}$ is the firing threshold, and $\tau$ is the membrane time constant. $H[t]$ denotes the pre-spike membrane potential obtained by leaky integration of $V[t-1]$ and $X[t]$. The output spike $S[t] \in \{0, 1\}$ is generated by the Heaviside step function $\text{Heav}(\cdot)$, which emits 1 when $H[t] \geq V_{\text{th}}$. The post-spike membrane potential $V[t]$ equals $H[t]$ if no spike occurs, and is reset to $V_{\text{reset}}$ otherwise. The time constant $\tau$ is treated as a learnable parameter, allowing the neuron's temporal dynamics to be optimized during training.

**Overall Architecture** DiceFormer consists of five main components: an Initialization Block, Projection Conv

Blocks, the SADA and SDA, and a final Classification Head (CH). Given an input spectrogram $I \in \mathbb{R}^{T \times C \times H \times W}$[1], the processing pipeline is as follows. The **Initialization** Block maps channels $C \to E$ and downsamples the spatial resolution by a factor of 4 ($H \to H/4$, $W \to W/4$). The first **Projection Block 1** further maps channels $E \to D_1$ and applies an additional $\times 2$ downsampling ($H/4 \to H/8$, $W/4 \to W/8$). The **SADA** module then splits the $D_1$ channels into two parallel streams of size $D_1/2$, where frequency attention operates over $N_{\text{freq}} = H/8$ tokens and temporal attention over $N_{\text{temp}} = W/8$ tokens. After this stage, the second **Projection Block 2** performs one more downsampling, changing feature map channels from $D_1 \to D_2$, and the **SDA** module applies unified self-attention on the flattened token sequence $N_{HW} = (H/16) \times (W/16)$ with $D_2$ feature channels. Finally, the SDA output is aggregated by Global Average Pooling (GAP) and passed to the **Classification Head (CH)** for prediction. Implementation details for each module are provided in Appendix A.

### 3.1. Initialization Block

The Initialization Block serves as the CNN stem (Liu et al., 2022) of DiceFormer, performing initial feature extraction and spatial downsampling from the input spectrogram. As shown in Figure 1, it consists of three sequential sub-blocks

---

[1]For static 2D inputs $I_s \in \mathbb{R}^{C \times H \times W}$, we replicate the frame $T$ times to form a length-$T$ sequence. The temporal axis $T$ is handled explicitly in spiking neuron layers, and is otherwise typically folded into the batch dimension.

(**Init-1**, **Init-2**, **Init-3**) and a parallel skip connection (**Skip**). **Init-1** applies a $7 \times 7$ Conv2d, BatchNorm, and MaxPool, followed by a Spiking Neuron (SN). The main path then proceeds through **Init-2**, which applies a $3 \times 3$ Conv2d, BatchNorm, and another MaxPool, again followed by an SN. The final stage, **Init-3**, applies a $3 \times 3$ Conv2d and BatchNorm. In parallel, the **Skip** path—branched from the input of **Init-2**—contains its own $3 \times 3$ Conv2d and BatchNorm to align channel dimensions with those of **Init-3**. The outputs of **Init-3** and **Skip** are then combined by element-wise addition, and the summed feature is passed through an SN. This output, $X_1$, has each spatial dimension reduced by a factor of 4 ($H \times W \rightarrow H/4 \times W/4$) and its channel dimension projected to the base embedding size $E$.

### 3.2. Projection Block

The Projection Conv Blocks act as transition layers (Huang et al., 2017) in the hierarchical architecture of DiceFormer. Their role is to further downsample feature maps while projecting channels to match the dimensionality required by subsequent attention modules (SADA and SDA). As illustrated in Figure 1, each block consists of two sub-blocks, **Project-1** and **Project-2**. **Project-1** performs downsampling with a $3 \times 3$ Conv2d, BatchNorm, and MaxPool (stride 2), thereby reducing each spatial dimension by half. Its output is then passed through an SN, which provides the input to **Project-2**. The latter applies a $3 \times 3$ Conv2d and BatchNorm without further altering spatial resolution. A skip connection adds the output of **Project-1** (before the SN) to the output of **Project-2**. The combined feature is finally processed by an SN to produce the block output.

### 3.3. Dice-Based Spike Attention

The self-attention mechanism in ANN-based Transformers relies heavily on Softmax normalization of real-valued feature vectors (Vaswani et al., 2017). In SNNs, however, information is represented as event-based binary spikes, making Softmax-based attention difficult to apply and ill-suited to the sparse, event-driven nature of spike signals (Zhou et al., 2023). Motivated by this mismatch, we revisit overlap-based spike attention and propose a density-aware formulation tailored to spiking computation.

Many spike self-attention variants replace Softmax-based dot-product attention with spike-compatible *overlap-only* scores. However, overlap-only scoring does not explicitly account for spike density, and can therefore favor high firing-rate queries/keys even without better alignment. To make this limitation precise, we first introduce basic overlap statistics.

**Definition 3.1** (Spike overlap statistics)**.** Let $q, k \in \{0, 1\}^{1 \times d}$ be binary spike *row* vectors. Define the spike counts $C(q) := \|q\|_1$ and $C(k) := \|k\|_1$. We define the

overlap

$$I(q, k) := \|q \odot k\|_1 = \sum_{\ell=1}^{d} q_\ell k_\ell, \qquad (4)$$

and the total activity $S(q, k) := C(q) + C(k)$.

**Standard Dot Product and Density Bias.** In standard spike-driven transformers, the attention score is typically derived from the dot product overlap $I(q, k)$. For query and key matrices $\mathbf{Q}, \mathbf{K}$, this results in a similarity map where each entry is $(\mathbf{Q}\mathbf{K}^\top)_{ij} = I(q_i, k_j)$. However, under a simple null model where $k$ has $C(k)$ active entries uniformly distributed over $d$ dimensions, the expected overlap scales as

$$\mathbb{E}[I(q, k)] = \frac{C(q)\,C(k)}{d}. \qquad (5)$$

This implies an inherent bias: dense spiking patterns yield higher attention scores regardless of actual feature alignment (see Appendix B.1 for derivation). Consequently, standard attention mechanisms are susceptible to domination by high-frequency noise.

**Our Approach: Element-wise Dice Attention.** To address the density bias, we propose **Spike Dice Similarity (SDS)**. This metric adapts the Dice coefficient to explicitly normalize overlap by total activity. Crucially, to avoid quadratic complexity, we compute similarity strictly between corresponding tokens (row-wise) via element-wise operations.

**Formulation.** Let $\mathbf{Q}, \mathbf{K} \in \{0, 1\}^{N \times d}$ be the query and key matrices ($N$: tokens, $d$: feature dim). We denote the $i$-th row vectors as $\mathbf{q}_i$ and $\mathbf{k}_i$, respectively. By applying the statistics from Definition 3.1 to each token pair $(\mathbf{q}_i, \mathbf{k}_i)$ and summing over the channel dimension $d$, we obtain the scalar overlap and activity values for each token.

First, we introduce a stability-adjusted inverse activity factor (with $\epsilon = 10^{-6}$):

$$c(\mathbf{q}_i, \mathbf{k}_i) := (S(\mathbf{q}_i, \mathbf{k}_i) + \epsilon)^{-1}. \qquad (6)$$

The element-wise SDS score $z_i \in \mathbb{R}$ is then defined as:

$$z_i = \mathrm{SDS}(\mathbf{q}_i, \mathbf{k}_i) := 2I(\mathbf{q}_i, \mathbf{k}_i)\, c(\mathbf{q}_i, \mathbf{k}_i). \qquad (7)$$

This formulation redefines similarity based on discrete signal activity ($\ell_1$ accumulation), treating spike counts as the energy unit. Unlike standard dot-product-based SNNs, SDS effectively suppresses density bias by normalizing overlap against this activity (see Appendix B.2). Furthermore, this approach leverages the binary nature of spikes to perform efficient computation, relying primarily on accumulation and avoiding computationally expensive square roots. Further theoretical context and comparisons with other metrics (e.g., F1, Jaccard) are detailed in Appendix C.

**Hardware-Efficient Linearization (Lin-SDS).** To facilitate efficient deployment on neuromorphic hardware constraints, we introduce **Lin-SDS**, a linearized variant derived via a first-order Taylor expansion (see Appendix D). This formulation adapts the density-aware metric into an affine transformation:

$$\text{SDS}_{\text{Lin}}(\mathbf{q}_i, \mathbf{k}_i) := I(\mathbf{q}_i, \mathbf{k}_i) - \lambda\big(C(\mathbf{q}_i) + C(\mathbf{k}_i)\big), \quad (8)$$

where $\lambda$ is a learnable parameter. By constraining operations to fixed-point addition and multiplication, Lin-SDS maximizes hardware compatibility while maintaining the theoretical benefits of density-aware scoring.

### 3.4. Spike Audio Dice Attention

The SADA module is designed to learn rich frequency-temporal features by explicitly decoupling and then fusing information from the frequency and temporal axes. The module's operation consists of three main stages.

**1. Input Processing and Decoupling.** Let $H_f := \frac{H}{8}$ and $W_t := \frac{W}{8}$ denote the frequency and temporal resolutions after patching, and let $D_1' := \frac{D_1}{2}$. We denote the output of the preceding projection conv block as

$$X_{\text{in}} \in \mathbb{R}^{T \times D_1 \times H_f \times W_t}. \quad (9)$$

After passing through a spiking neuron (SN), we obtain

$$X_{\text{spike}} \in \mathbb{R}^{T \times D_1 \times H_f \times W_t}, \quad (10)$$

which is split along the channel dimension into two halves and routed to a frequency branch and a temporal branch in parallel. The original $X_{\text{in}}$ is retained for the final residual connection after feature fusion. In the frequency branch, the temporal axis ($W_t$) is merged into the batch dimension so that each temporal slice forms an independent sequence of length $N_{\text{freq}} = H_f$. In the temporal branch, the frequency axis ($H_f$) is merged into the batch dimension so that each frequency slice forms an independent sequence of length $N_{\text{temp}} = W_t$.

**2. Parallel Attention Streams.** The two halves of the split tensor, denoted as $X$, are processed in parallel: one stream operates along the frequency axis and the other along the temporal axis. We reshape them as

$$\begin{aligned} \text{Freq} &\in \mathbb{R}^{T \times N_{\text{freq}} \times D_1'}, \quad N_{\text{freq}} = H_f, \\ \text{Temp} &\in \mathbb{R}^{T \times N_{\text{temp}} \times D_1'}, \quad N_{\text{temp}} = W_t. \end{aligned} \quad (11)$$

From each reshaped tensor, query ($Q$), key ($K$), and value ($V$) spike vectors are generated using parallel Linear-BatchNorm-SN branches:

$$\begin{aligned} Q &= \text{SN}\big(\text{BN}(XW_Q)\big), \\ K &= \text{SN}\big(\text{BN}(XW_K)\big), \\ V &= \text{SN}\big(\text{BN}(XW_V)\big). \end{aligned} \quad (12)$$

The attention output is computed by forming spike-dice weights from the SDS between $Q$ and $K$. This continuous-valued score is passed through an SN to obtain a binary map, which is then applied to $V$ via Hadamard product with channel-wise broadcasting:

$$\text{Attn} = \text{SN}\big(\text{SDS}(Q, K)\big) \odot V. \quad (13)$$

Applying this procedure separately yields the frequency-focused output $\text{Attn}_{\text{freq}}$ and the temporal-focused output $\text{Attn}_{\text{temp}}$.

**3. Feature Fusion and Output.** The frequency- and temporal-attention outputs are reshaped to restore the original spatial dimensions and concatenated along the channel dimension to reconstruct the size $D_1$. The combined tensor is processed by a pointwise convolution (PWConv) (Howard et al., 2017) with BatchNorm to learn cross-dependencies. A residual connection with $X_{\text{in}}$ is then applied, followed by an SN and an MLP block:

$$\begin{aligned} X_{\text{fused}} &= \text{BN}\Big(\text{PWConv}\big(\text{concat}(\text{Attn}_{\text{freq}}, \text{Attn}_{\text{temp}})\big)\Big) \\ &\quad + X_{\text{in}}, \\ X_{\text{out}} &= \text{MLP}\big(\text{SN}(X_{\text{fused}})\big). \end{aligned} \quad (14)$$

### 3.5. Spike Dice Attention

Whereas the SADA module explicitly decouples frequency and temporal information, the SDA performs unified attention over the combined frequency–temporal dimensions.

Let $H_s := \frac{H}{16}$ and $W_s := \frac{W}{16}$. Given an input tensor

$$X_{\text{in}} \in \mathbb{R}^{T \times D_2 \times H_s \times W_s}, \quad (15)$$

from the second Projection Conv block, we flatten the spatial dimensions into a sequence of length

$$N_{HW} := H_s W_s, \quad (16)$$

yielding $X \in \mathbb{R}^{T \times N_{HW} \times D_2}$. After passing through a spiking neuron (SN), we obtain

$$X_{\text{spike}} \in \mathbb{R}^{T \times N_{HW} \times D_2}. \quad (17)$$

From $X_{\text{spike}}$, query ($Q$), key ($K$), and value ($V$) vectors are generated via parallel Linear-BatchNorm-SN branches:

$$\begin{aligned} Q &= \text{SN}\big(\text{BN}(X_{\text{spike}}W_Q)\big), \\ K &= \text{SN}\big(\text{BN}(X_{\text{spike}}W_K)\big), \\ V &= \text{SN}\big(\text{BN}(X_{\text{spike}}W_V)\big). \end{aligned} \quad (18)$$

Attention is computed using the SDS. The resulting similarity map is passed through an SN to yield binary spike-dice

*Table 1.* Performance comparison across datasets. Scores are reported as mAP for AudioSet (20k) and accuracy (%) for ESC-50 and SCV2. We evaluated three model variants: DiceFormer-10-S, DiceFormer-10-M, and DiceFormer-10-L. Here, "10" denotes a total of 10 attention layers, consisting of 5 SADA and 5 SDA layers. The S, M, and L suffixes distinguish the models by their attention channel dimensions, which are set to [192, 384] for S, [256, 512] for M, and [384, 768] for L, respectively.

| Model | Type | Direct training | Parameters (M) | Energy (mJ) | Time step | Score |
|---|---|---|---|---|---|---|
| **AudioSet (20k)** — mAP | | | | | | |
| AST (Gong et al., 2021) | ANN | - | 88.1 | 475.64 | - | 0.148 |
| SSAST-S (Gong et al., 2022) | ANN | - | 23 | 176.82 | - | 0.165 |
| DTF-AT (Alex et al., 2024) | ANN | - | 69 | 153.18 | - | 0.187 |
| Spikformer (Zhou et al., 2023) | SNN | ✓ | 65.9 | 18.82 | 4 | 0.136 |
| Spike-driven Transformer (Yao et al., 2023a) | SNN | ✓ | 65.9 | 8.15 | 4 | 0.130 |
| Spike-driven Transformer V2 (Yao et al., 2024) | SNN | ✓ | 55.0 | 14.92 | 4 | 0.138 |
| QKFormer (Zhou et al., 2024) | SNN | ✓ | 64.5 | 43.43 | 4 | 0.147 |
| DiceFormer-10-S (Ours) | SNN | ✓ | 13.7 | 5.34 | 4 | 0.145 |
| DiceFormer-10-M (Ours) | SNN | ✓ | 24.2 | 9.55 | 4 | 0.157 |
| DiceFormer-10-L (Ours) | SNN | ✓ | 54.3 | 6.18 | 1 | 0.153 |
| DiceFormer-10-L (Ours) | SNN | ✓ | 54.3 | 17.80 | 4 | **0.161** |
| **ESC-50** — Acc (%) | | | | | | |
| AST (Gong et al., 2021) | ANN | - | 87.2 | 260.54 | - | 88.7* |
| SSAST-S (Gong et al., 2022) | ANN | - | 23 | 72.63 | - | 85.4* |
| DTF-AT (Alex et al., 2024) | ANN | - | 68.6 | 77.28 | - | 89.19* |
| DiceFormer-10-S (Ours) | SNN | ✓ | 13.5 | 5.17 | 4 | 85.37 |
| DiceFormer-10-M (Ours) | SNN | ✓ | 24.0 | 7.95 | 4 | 85.47 |
| **SCV2** — Acc (%) | | | | | | |
| AST (Gong et al., 2021) | ANN | - | 86.9 | 44.11 | - | 98.11* |
| SSAST-S (Gong et al., 2022) | ANN | - | 23 | 11.32 | - | 97.70* |
| DTF-AT (Alex et al., 2024) | ANN | - | 68.6 | 19.32 | - | 98.30* |
| DCLS-Delays (Hammouamri et al., 2024) | SNN | ✓ | 2.5 | - | - | 95.35 |
| SIDC-KWS (Lim & Kim, 2025b) | SNN | ✓ | 0.4 | - | 8 | 94.70 |
| DiceFormer-10-S (Ours) | SNN | ✓ | 13.5 | 5.31 | 4 | **97.27** |

\* Pre-trained on external datasets. Unmarked scores indicate training from scratch.

weights, which are then applied to $V$ via Hadamard product with broadcasting:

$$\text{Attn} = \text{SN}\big(\text{SDS}(Q, K)\big) \odot V. \tag{19}$$

Finally, $\text{Attn}$ is reshaped back to the spatial view, passed through a pointwise convolution (PWConv) with Batch-Norm, combined with the residual input $X_{\text{in}}$, and processed by an SN and an MLP to produce the final output:

$$\begin{aligned} X_{\text{fused}} &= \text{BN}\big(\text{PWConv}(\text{Attn})\big) + X_{\text{in}}, \\ X_{\text{out}} &= \text{MLP}\big(\text{SN}(X_{\text{fused}})\big). \end{aligned} \tag{20}$$

To instantiate the hardware-efficient **Lin-SDA**, the SDS metric is simply replaced by the Lin-SDS formulation (Eq. 8). Both SADA and SDA are naturally extended to multi-head variants, with detailed formulations provided in Appendix E.

## 4. Experiments

We conduct extensive experiments to evaluate the effectiveness of DiceFormer across three widely used benchmark datasets: AudioSet (20k) (Gemmeke et al., 2017) and ESC-50 (Piczak, 2015) for general audio classification, and Speech Commands V2 (SCV2) (Warden, 2018) for keyword spotting. These datasets collectively cover large-scale, environmental, and speech-focused tasks, providing a comprehensive assessment of model generalizability. All Dice-Former variants are trained from scratch to ensure that performance gains are achieved without reliance on pre-training or external supervision. Specifically, we adopt direct training (Wu et al., 2019) with surrogate gradients (Neftci et al., 2019; Fang et al., 2021). Comprehensive training protocols and hyperparameters are provided in Appendix F, with energy calculation methodology in Appendix G and training/inference times in Appendix H. Table 1 reports the main

*Table 2.* Combined ablation studies on the AudioSet-20k dataset. The **first section** analyzes attention components and ordering. The **second section** evaluates binarization. The **third section** examines the hardware-efficient Lin-SDA variant across model scales. The **fourth section** demonstrates the generalization of SDA on existing architectures.

| Methods | Time step | SADA:SDA ratio | mAP |
|---|---|---|---|
| ANALYSIS OF ATTENTION COMPONENTS | | | |
| DiceFormer-10-L (SDSA (Yao et al., 2023a) only) | 4 | 5:5 | 0.126 |
| DiceFormer-10-L (SSA (Zhou et al., 2023) only) | 4 | 5:5 | 0.142 |
| DiceFormer-10-L (SADA only) | 4 | 5:5 | 0.157 |
| DiceFormer-10-L (SDA only) | 4 | 5:5 | 0.158 |
| DiceFormer-10-L (SDA → SADA) | 4 | 5:5 | 0.157 |
| DiceFormer-10-L (SADA → SDA) | 4 | 6:4 | 0.160 |
| DiceFormer-10-L (SADA → SDA) | 4 | 4:6 | 0.155 |
| DiceFormer-10-L (SADA → SDA, Full Model) | 4 | 5:5 | **0.161** |
| EFFECTIVENESS OF BINARIZED ATTENTION MAP | | | |
| DiceFormer-10-L (SDS-continuous) | 4 | 5:5 | 0.160 |
| DiceFormer-10-L (SDS-binarized) | 4 | 5:5 | **0.161** |
| EFFECTIVENESS OF LIN-SDA | | | |
| DiceFormer-10-S (Lin-SDA) | 4 | 5:5 | 0.143 |
| DiceFormer-10-M (Lin-SDA) | 4 | 5:5 | 0.154 |
| DiceFormer-10-L (Lin-SDA) | 4 | 5:5 | 0.159 |
| GENERALIZATION OF SDA | | | |
| Spike-driven Transformer (baseline) | 4 | - | 0.130 |
| Spike-driven Transformer (+SDA) | 4 | - | **0.142 (+0.012)** |
| Spike-driven Transformer V2 (baseline) | 4 | - | 0.138 |
| Spike-driven Transformer V2 (+SDA) | 4 | - | **0.144 (+0.006)** |
| QKFormer (baseline) | 4 | - | 0.147 |
| QKFormer (+SDA) | 4 | - | **0.149 (+0.002)** |

*Table 3.* Drop-in SDA replacement on CIFAR-100. The values in parentheses indicate performance improvements over the original attention mechanisms.

| Model | Attn. | Top-1 (%) |
|---|---|---|
| Spike-driven Transformer | original | 78.40 |
| (Yao et al., 2023a) | SDA (ours) | **78.99 (+0.59)** |
| QKFormer | original | 81.15 |
| (Zhou et al., 2024) | SDA (ours) | **81.41 (+0.26)** |

comparison results used throughout the paper. Detailed multi-run statistics, including means, standard deviations, and 95% confidence intervals for each dataset, are reported in Appendix I.

### 4.1. Performance Comparison

As shown in Table 1, **DiceFormer-10-L** establishes a new SNN SOTA on **AudioSet (20k)** with **0.161 mAP**, outperforming prior SNNs and, crucially, surpassing the **scratch-trained** ANN baseline AST (0.148 mAP) by a clear margin. The lightweight **DiceFormer-10-S** (0.145 mAP) further highlights our architecture's efficiency, effectively performing comparably to the previous SNN SOTA (QKFormer) while requiring only a fraction of the parameters and energy.

This efficiency advantage extends to downstream tasks. On **ESC-50**, DiceFormer-10-M achieves **85.47%** accuracy while consuming only **7.95 mJ**. Similarly, on **SCV2**, DiceFormer-10-S reaches **97.27%** using only **5.31 mJ**, demonstrating performance comparable to ANN baselines with significantly lower energy consumption.

### 4.2. Ablation Study

We conduct ablation studies on **AudioSet-20k** using **DiceFormer-10-L** ($T=4$). As detailed in Table 2, we analyze: (i) ATTENTION COMPONENTS against prior baselines; (ii) the impact of MAP BINARIZATION; (iii) the efficiency of LIN-SDA; and (iv) the GENERALIZATION OF SDA on existing backbones. We further verify applicability to vision

tasks on **CIFAR-100** (Table 3).

**Analysis of Components & Structure.** Table 2 confirms the superiority of our density-aware modules (**SADA, SDA**) over unaware baselines (SSA, SDSA), boosting mAP from ∼0.14 to over 0.157. The optimal configuration (**SADA → SDA**, 5:5 ratio) achieves **0.161 mAP**, demonstrating clear synergy. **SDA** also proves versatile as a drop-in replacement, consistently improving existing SNNs (e.g., **+0.006** on SDT V2), while the binarized attention map maintains full floating-point accuracy (**0.161 mAP**), validating the fully spike-based design.

**Effectiveness of Lin-SDA.** We evaluate the performance trade-off of **Lin-SDA**, which explicitly applies the **Lin-SDS** metric (Eq. 8) to our SDA mechanism. As shown in Table 2, employing this formulation incurs a negligible performance drop (achieving **0.159 mAP**) compared to the standard SDA. This confirms that Lin-SDA successfully maintains the theoretical robustness of Dice-based scoring while **further streamlining computations for neuromorphic hardware constraints**, validating its suitability for practical deployment.

**Preliminary Vision-Domain Assessment.** We also evaluate SDA on **CIFAR-100** (Table 3) as a preliminary assessment of its compatibility with image inputs. Replacing the original attention modules with SDA yields modest accuracy improvements in Spike-driven Transformer and QKFormer (**+0.59 pp** and **+0.26 pp**, respectively). These results indicate that, on this benchmark, SDA is compatible with existing spiking vision Transformer backbones.

### 4.3. Correlation Between Attention Score and Spike Density

To quantify the coupling between attention scores and firing activity, we measured the Pearson correlation between attention weights and input spike density on AudioSet-20k (layer-wise details in Appendix J). Our analysis reveals that prior methods exhibit strong density coupling (e.g., Spikformer: 0.8612), whereas DiceFormer variants show significantly weaker coupling (L: 0.082). We observe a strong negative correlation ($r = -0.805$) between this density coupling and mAP (Figure 2), suggesting that decoupling attention from simple firing rates can be considered a contributing factor to the improved performance.

## 5. Hardware Implementation and Efficiency Analysis

**Neuromorphic Hardware Primitives.** Our proposed **SDA mechanism** is designed to replace expensive floating-point matrix multiplications with efficient bitwise logic (detailed in Appendix K). Specifically, the overlap $I(\mathbf{q}_i, \mathbf{k}_i)$ and total activity $S(\mathbf{q}_i, \mathbf{k}_i)$ are computed using only bitwise AND,

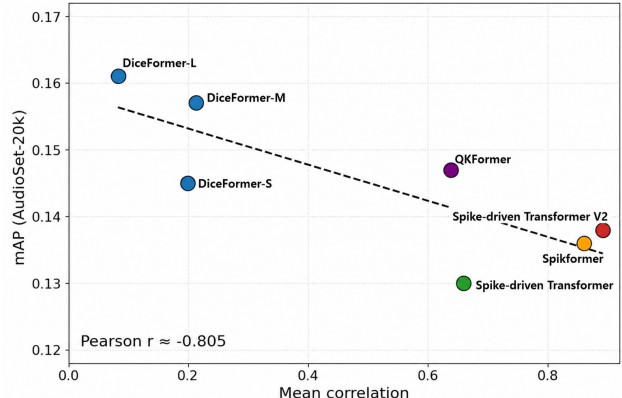

*Figure 2.* Relationship between the model-wise mean score–density correlation and AudioSet-20k mAP.

*Table 4.* Runtime comparison. Average latency is reported as mean ± standard deviation (ms).

| Method | Latency (ms) | Complexity |
|---|---|---|
| Spikformer (Zhou et al., 2023) | $2.95 \pm 1.15$ | $\mathcal{O}(ND^2)$ |
| SDT V1 (Yao et al., 2023a) | $1.70 \pm 0.05$ | $\mathcal{O}(ND)$ |
| SDT V2 (Yao et al., 2024) | $4.68 \pm 8.72$ | $\mathcal{O}(ND^2)$ |
| DiceFormer (Ours) | $1.67 \pm 0.03$ | $\mathcal{O}(ND)$ |
| DiceFormer (Lin-SDA, Ours) | $1.60 \pm 0.08$ | $\mathcal{O}(ND)$ |

popcount, and addition. Regarding the inverse activity factor $c(\cdot)$, since the total spike count is strictly bounded within $2d$, it is efficiently implemented via a Look-Up Table (LUT). Alternatively, we provide **Lin-SDA** as a computation-efficient variant, ensuring that the entire process relies solely on simple integer arithmetic and logic.

**Runtime Analysis.** As shown in Table 4, prior methods (Spikformer, SDT V2) incur high costs due to quadratic channel scaling ($\mathcal{O}(ND^2)$), yielding slower latencies (2.95–4.68 ms). In contrast, DiceFormer leverages linear complexity ($\mathcal{O}(ND)$) to reduce latency to 1.67 ms, with **Lin-SDA achieving the lowest at 1.60 ms**. This result confirms that Lin-SDA further maximizes the efficiency of our linear framework, validating its suitability for resource-constrained neuromorphic deployment.

## 6. Conclusion

In this work, we address two key challenges in spike-based Transformers: the *density bias* in spike attention and the limited exploration of SNN Transformers for audio. We introduce **DiceFormer**, which combines **SDA** to mitigate density bias with **SADA** to capture audio-specific inductive biases. Supported by theoretical and empirical analyses, DiceFormer achieves new state-of-the-art performance among SNNs on AudioSet-20k and ESC-50, while remaining competitive on SCV2. Although our primary focus is audio, our CIFAR-100 experiments provide a prelimi-

nary assessment of SDA's compatibility with spiking vision Transformer backbones. We also propose **Lin-SDA**, a computation-efficient variant that preserves performance comparable to SDA. We hope our findings encourage further exploration of density-aware attention mechanisms in neuromorphic models.

**Future Work.** This work focuses on a from-scratch training setting to enable a clean comparison against prior SNN methods. Building on these strong baselines, we plan to explore two main directions: (1) deploying Lin-SDA on physical neuromorphic hardware to validate real-world energy efficiency, and (2) investigating large-scale pre-training strategies to further scale up model capacity and align with modern training paradigms.

## Acknowledgements

This work was supported by the National Research Foundation of Korea (NRF) grants funded by the Korean Government under Grants RS-2024-00422599, RS-2026-25492338, and RS-2025-23963845.

## Impact Statement

This work aims to advance efficient SNN-based audio classification by addressing density bias in spiking attention and improving the energy–accuracy trade-off. Its potential positive impacts include reducing the computational and energy costs of audio classification models, which may support more efficient deployment on resource-constrained hardware. As with other audio classification methods, potential negative impacts may arise if such technologies are used in privacy-sensitive acoustic monitoring or surveillance settings without appropriate consent, transparency, and safeguards. Our study focuses on algorithmic development and evaluation on standard benchmarks, and does not introduce a deployed system or collect new user data. We encourage responsible use that considers privacy, fairness across acoustic environments, and potential misuse.

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

*Table 5.* Detailed architecture of DiceFormer. 'k', 's', and 'p' denote kernel size, stride, and padding, respectively. The model has three variants: **DiceFormer-10-S** ($E = 96, D_1 = 192, D_2 = 384$), **DiceFormer-10-M** ($E = 128, D_1 = 256, D_2 = 512$), and **DiceFormer-10-L** ($E = 192, D_1 = 384, D_2 = 768$). For all variants, the SADA and SDA modules use 2 and 8 attention heads, respectively.

| Stage | Layer Name | Key Operations | # Tokens | Channels | | |
|---|---|---|---|---|---|---|
| | | | | DiceFormer-10-S | DiceFormer-10-M | DiceFormer-10-L |
| 1 | Initialization Block | Init-1:
  Conv2d(k=7,s=1,p=3)
  MaxPool(k=2,s=2)
Init-2:
  Conv2d(k=3,s=1,p=1)
  MaxPool(k=2,s=2)
Init-3:
  Conv2d(k=3,s=1,p=1)
Skip:
  Conv2d(k=3,s=2,p=1) | $\frac{H}{4} \times \frac{W}{4}$ | 96 | 128 | 192 |
| 2 | Projection Block 1 | Project-1:
  Conv2d(k=3,s=1,p=1)
  MaxPool(k=2,s=2)
Project-2:
  Conv2d(k=3,s=1,p=1) | $\frac{H}{8} \times \frac{W}{8}$ | 192 | 256 | 384 |
| 3 | SADA Module | – *Frequency Stream* –
Freq-Attention (2 Heads) | $\frac{H}{8} \times \frac{W}{8}$ | 96 | 128 | 192 |
| | | – *Temporal Stream* –
Temp-Attention (2 Heads) | $\frac{H}{8} \times \frac{W}{8}$ | 96 | 128 | 192 |
| | | PWConv & MLP | $\frac{H}{8} \times \frac{W}{8}$ | 192 | 256 | 384 |
| 4 | Projection Block 2 | Project-1:
  Conv2d(k=3,s=1,p=1)
  MaxPool(k=2,s=2)
Project-2:
  Conv2d(k=3,s=1,p=1) | $\frac{H}{16} \times \frac{W}{16}$ | 384 | 512 | 768 |
| 5 | SDA Module | Unified Spike Dice Attention (8 Heads)
Fusion (PWConv) & MLP | $\frac{H}{16} \times \frac{W}{16}$
$\frac{H}{16} \times \frac{W}{16}$ | 384
384 | 512
512 | 768
768 |
| 6 | Classification Head | Global Average Pooling, Linear | – | - | | |

# A. DiceFormer Implementation Details

We configure three versions of the DiceFormer model based on the attention channel dimensions: DiceFormer-10-S, DiceFormer-10-M, and DiceFormer-10-L. The number 10 in the model names denotes the total count of SADA and SDA blocks, with all versions containing a total of 10 such blocks. The detailed architecture for each model is summarized in **Table 5**.

The model processes an input tensor with a shape of $T \times C \times H \times W$, where $T, C, H,$ and $W$ denote the time step, channels, height, and width, respectively. For simplicity, the batch dimension is omitted from all tensor shapes described below.

# B. Density Bias of Overlap-Only Spike Attention and Dice-Based Spike Attention

**B.1. Density bias of overlap-only scores: deriving** $\mathbb{E}[I(q, k)] = \frac{C(q)C(k)}{d}$

In this section, we derive the expected overlap in Eq. (5) under a simple null model and show that overlap-only scores are inherently biased by spike density.

**(Assumption) Null model: uniform distribution with a fixed spike count.** Let $q \in \{0, 1\}^{1 \times d}$ be a fixed binary spike query with $C(q) = \|q\|_1$. Assume that the key $k \in \{0, 1\}^{1 \times d}$ contains **exactly** $C(k)$ ones, whose active positions are

selected **uniformly** among the $d$ dimensions (i.e., selecting $C(k)$ distinct indices without replacement). The overlap is defined as

$$I(q, k) = \sum_{\ell=1}^{d} q_\ell k_\ell. \tag{21}$$

**(Derivation) A calculation via linearity of expectation.** Define $X_\ell := q_\ell k_\ell \in \{0, 1\}$ for each dimension $\ell$. Then

$$I(q, k) = \sum_{\ell=1}^{d} X_\ell, \qquad \mathbb{E}[I(q, k)] = \sum_{\ell=1}^{d} \mathbb{E}[X_\ell]. \tag{22}$$

Since $q$ is fixed, $\mathbb{E}[X_\ell] = q_\ell \mathbb{E}[k_\ell]$. Under the uniform-selection assumption, the probability that an arbitrary coordinate $\ell$ is active is

$$\mathbb{P}(k_\ell = 1) = \frac{C(k)}{d} \quad \Rightarrow \quad \mathbb{E}[k_\ell] = \frac{C(k)}{d}. \tag{23}$$

Therefore,

$$\mathbb{E}[I(q, k)] = \sum_{\ell=1}^{d} q_\ell \frac{C(k)}{d} = \frac{C(k)}{d} \sum_{\ell=1}^{d} q_\ell = \frac{C(q)C(k)}{d}. \tag{24}$$

Hence,

$$\boxed{\mathbb{E}[I(q, k)] = \frac{C(q)\, C(k)}{d}} \tag{25}$$

which matches Eq. (5).

**(Interpretation) What this density bias implies.** The above result shows that **even under a null model with no alignment information**, the expected overlap increases proportionally to $C(q)$ and $C(k)$. In particular, for a fixed $q$, if two key candidates $k^{(1)}, k^{(2)}$ satisfy $C(k^{(1)}) > C(k^{(2)})$, then

$$\mathbb{E}[I(q, k^{(1)})] - \mathbb{E}[I(q, k^{(2)})] = \frac{C(q)}{d} \big( C(k^{(1)}) - C(k^{(2)}) \big) > 0, \tag{26}$$

implying that overlap-only logits structurally favor higher firing-rate (denser) keys regardless of intrinsic similarity. This motivates a density-aware scoring rule in SNN attention, where spike density can vary substantially across tokens, channels, and time.

### B.2. Dice normalization mitigates density bias under the null model

Under the same assumption as Appendix B.1, we compute the expectation of the Dice-normalized score and show that the density bias of overlap-only scoring is mitigated. Using the multiplication-form notation from Eq. 7, and considering a single pair of query and key vectors $\mathbf{q}, \mathbf{k} \in \{0, 1\}^d$ for simplicity,

$$\mathrm{SDS}(q, k) = \big(2I(q, k)\big) c(q, k), \qquad c(q, k) := \big(C(q) + C(k) + \epsilon\big)^{-1}. \tag{27}$$

**(Derivation) A calculation via linearity of expectation.** Fixing $q$ and conditioning on a given spike count $C(k)$, $c(q, k)$ is a constant. Hence, by linearity of expectation,

$$\mathbb{E}\big[\mathrm{SDS}(q, k)\big] = \mathbb{E}\Big[\big(2I(q, k)\big) c(q, k)\Big] = 2\, \mathbb{E}\big[I(q, k)\big]\, c(q, k). \tag{28}$$

Substituting Appendix B.1's result $\mathbb{E}[I(q, k)] = \frac{C(q)C(k)}{d}$ yields

$$\mathbb{E}\big[\mathrm{SDS}(q, k)\big] = \frac{2}{d} C(q)\, C(k)\, c(q, k). \tag{29}$$

**(Interpretation) What this implies for density bias.** While overlap-only scoring satisfies $\mathbb{E}[I(q,k)] \propto C(k)$ and thus increases linearly with the key density, Dice normalization introduces the factor $c(q,k)$, which decreases as $C(k)$ grows. As a result,

$$\mathbb{E}\big[\mathrm{SDS}(q,k)\big] = \frac{2}{d}\, C(q)\, C(k)\, c(q,k) \tag{30}$$

exhibits diminishing returns with respect to $C(k)$, mitigating the structural advantage of higher-density keys under the null model.

## C. Justification for Choosing Dice Similarity

Our goal is to mitigate the *density bias* of overlap-only spike attention, where the raw overlap $I(q,k)$ can increase simply because the marginal activities $C(q)$ and $C(k)$ are large, even without improved alignment. We compare candidate similarity measures under the following criteria: (i) an explicit normalization/compensation with respect to marginal activity (i.e., a density-aware set similarity), (ii) computational suitability for SNNs with a **computation-efficient** realization—preferably relying on spike-driven accumulation and enabling an efficient implementation of the normalization term and (iii) the ability to yield a simple score that preserves *relative ranking* (which attention primarily relies on) under an efficient approximation. Under these considerations, we adopt the Dice coefficient.

**1) F1 score.** For binary spikes $q, k \in \{0,1\}^d$, when viewed as sets of active (1-valued) coordinates, the Dice coefficient can be interpreted as an F1-type measure and is effectively equivalent in form under this set perspective. Thus, F1 and Dice share the same normalization principle—correcting overlap by total marginal activity. In this work, we use the Dice form because it is natural from a set-similarity viewpoint and structurally convenient: its Taylor expansion yields a simple approximated score.

**2) Cosine similarity.** For spikes, cosine similarity can be written as

$$\cos(q,k) = \frac{I(q,k)}{\sqrt{C(q)C(k)}}, \tag{31}$$

which normalizes the overlap by $\sqrt{C(q)C(k)}$ and can reduce density bias to some extent. However, cosine requires computing a square-root term in the denominator, which complicates a **computation-efficient implementation** compared to Dice (e.g., requiring a larger or more complex fixed-point approximation/LUT). Moreover, by the AM–GM inequality,

$$\frac{C(q)+C(k)}{2} \geq \sqrt{C(q)C(k)} \quad \Rightarrow \quad \mathrm{Dice}(q,k) = \frac{2I}{C(q)+C(k)} \leq \frac{I}{\sqrt{C(q)C(k)}} = \cos(q,k), \tag{32}$$

so for a fixed overlap $I$, Dice assigns a more conservative score than cosine. Since our objective is to suppress score inflation caused by increased activity, this behavior is consistent with the desired bias-mitigation property. Importantly, Dice uses the simple marginal term $S = C(q) + C(k)$, enabling an efficient reciprocal approximation (e.g., via a compact fixed-point LUT), and we additionally evaluate a first-order Taylor approximation of Dice as an ablation; deriving an equally simple ranking-preserving approximation is less straightforward for cosine due to the $\sqrt{C(q)C(k)}$ structure.

**3) Jaccard similarity (IoU).** Jaccard similarity is defined as

$$\mathrm{Jaccard}(q,k) = \frac{I(q,k)}{C(q)+C(k)-I(q,k)} = \frac{I}{S-I}. \tag{33}$$

As a classical set similarity, it normalizes the overlap by a union-like term and can partially correct density effects. However, unlike Dice's $2I/S$, Jaccard uses a denominator $S-I$ that depends explicitly on the overlap $I$, resulting in a more nonlinear structure where the normalization term varies with the overlap itself.

This dependence can reduce simplicity from the implementation and approximation perspectives. For instance, implementing $(S-I)^{-1}$ must account for the joint variation of $S$ and $I$, which may require indexing/approximating over a larger two-variable space (e.g., a larger LUT) or additional bookkeeping. Moreover, even with a first-order Taylor expansion, the resulting form can be less interpretable than Dice's marginal-only normalization.

In our setting, the key benefit of Dice is that its denominator depends only on marginal counts, i.e., $S = C(q) + C(k)$, which enables an efficient reciprocal approximation (e.g., via a compact fixed-point LUT) and also admits a simple first-order Taylor approximation that we evaluate as an ablation.

**4) Hamming distance.** Hamming distance counts coordinate-wise mismatches:

$$\text{Ham}(q, k) = \sum_{\ell} |q_\ell - k_\ell|. \tag{34}$$

Because it is based on coordinate-wise agreement/disagreement, $(0, 0)$ *inactive* agreements are implicitly treated as matches. However, SNN activations are often extremely sparse, so $(0, 0)$ agreements can dominate the distance/similarity signal. As a result, many keys can obtain very similar scores even when their informative active supports differ, reducing discriminability. This weak separation can induce an *attention collapse* behavior, where attention becomes overly uniform rather than focusing on informative keys. In contrast, Dice-based scoring (SDS) normalizes by marginal activity $C(q)$ and $C(k)$, reducing the influence of inactive coordinates and better reflecting overlap on active supports.

## D. Dice normalization and SDS via a first-order Taylor approximation

**First-order Taylor expansion of the Dice function.** Let

$$F(I, S) := \frac{2I}{S} \quad (S > 0), \tag{35}$$

and consider a first-order Taylor approximation around an expansion point $(I_0, S_0)$ with $S_0 > 0$. The partial derivatives are

$$\frac{\partial F}{\partial I} = \frac{2}{S}, \qquad \frac{\partial F}{\partial S} = -\frac{2I}{S^2}. \tag{36}$$

Therefore, the first-order approximation at $(I_0, S_0)$ is

$$F(I, S) \approx F(I_0, S_0) + \frac{2}{S_0}(I - I_0) - \frac{2I_0}{S_0^2}(S - S_0). \tag{37}$$

This can be rearranged into an affine form:

$$F(I, S) \approx \alpha(I - \lambda S) + \beta, \tag{38}$$

where $\alpha := 2/S_0$, $\lambda := I_0/S_0$, and $\beta := 2I_0/S_0$.

**Rank preservation and the Linearized SDS.** In attention mechanisms, the **relative ranking** of scores is typically what matters. Since $\alpha > 0$ and $\beta$ is constant, removing the positive scale $\alpha$ and constant shift $\beta$ preserves the ordering of candidates. Hence, under the **first-order Taylor approximation**, the key term determining the rank simplifies to

$$I - \lambda S = I - \lambda(C(q) + C(k)). \tag{39}$$

This derivation leads to our hardware-efficient surrogate, denoted as **Linearized SDS (Lin-SDS)**:

$$\boxed{\text{SDS}_{\text{Lin}}(q, k) := I(q, k) - \lambda(C(q) + C(k)).} \tag{40}$$

Finally, in our formulation, we treat $\lambda$ as a **learnable parameter constrained to be non-negative**, allowing the model to adapt automatically to the data distribution and network dynamics without expensive division operations.

## E. Spike Dice Attention: Multi-Head Formulation

### E.1. Multi-Head in SADA

The SADA module applies multi-head attention to both the frequency and temporal streams. In each stream, the input tensor has shape $T \times N \times D$ (with stream-specific $N$ and $D$). We split the feature dimension $D$ into $k$ heads, each with a per-head width of $d = D/k$, and reshape it to $T \times k \times N \times d$, enabling parallel attention across heads.

**Frequency branch (multi-head).** Given $Q_f, K_f, V_f \in \{0,1\}^{T \times N_{\text{freq}} \times D'_1}$ with $D'_1 = kd$, for each head $h \in \{1, \ldots, k\}$ we take $Q_f^{(h)}, K_f^{(h)}, V_f^{(h)} \in \{0,1\}^{T \times N_{\text{freq}} \times d}$ and compute

$$\text{Attn}_{\text{freq}}^{(h)} = \text{SN}(\text{SDS}(Q_f^{(h)}, K_f^{(h)})) \odot V_f^{(h)}.$$

The final frequency output is obtained by merging head-wise results along the head axis to recover the original width: $\text{Attn}_{\text{freq}} \in \{0,1\}^{T \times N_{\text{freq}} \times D'_1}$.

**Temporal branch (multi-head).** Analogously, with $Q_t, K_t, V_t \in \{0,1\}^{T \times N_{\text{temp}} \times D'_1}$ and $D'_1 = kd$, we compute

$$\text{Attn}_{\text{temp}}^{(h)} = \text{SN}(\text{SDS}(Q_t^{(h)}, K_t^{(h)})) \odot V_t^{(h)},$$

and merge along the head axis to obtain $\text{Attn}_{\text{temp}} \in \{0,1\}^{T \times N_{\text{temp}} \times D'_1}$.

**Feature Fusion and Output.** The outputs of the two streams, $\text{Attn}_{\text{freq}}$ and $\text{Attn}_{\text{temp}}$, are first **reshaped to restore their original spatial dimensions** and then concatenated along the channel dimension, restoring the original width $D_1$. The combined tensor is passed through a PWConv to fuse inter-dependencies, followed by a residual connection and an MLP:

$$X_{\text{cat}} = \text{concat}(\text{Attn}_{\text{freq}}, \text{Attn}_{\text{temp}}),$$
$$X_{\text{fused}} = \text{BN}(\text{PWConv}(X_{\text{cat}})) + X_{\text{in}},$$
$$X_{\text{out}} = \text{MLP}(\text{SN}(X_{\text{fused}})).$$

### E.2. Multi-Head in SDA (Unified)

The unified SDA also employs multi-head attention. The input tensor $T \times N_{HW} \times D_2$ is split into $k$ heads, yielding $d = D_2/k$ and the reshaped form $T \times k \times N_{HW} \times d$ for parallel processing.

**Unified branch (multi-head).** For $Q, K, V \in \{0,1\}^{T \times N_{HW} \times D_2}$ with $D_2 = kd$ and $N_{HW} = (H/16)(W/16)$, each head computes

$$\text{Attn}^{(h)} = \text{SN}(\text{SDS}(Q^{(h)}, K^{(h)})) \odot V^{(h)}.$$

Merging the $k$ head-wise outputs restores the width $D_2$: $\text{Attn} \in \{0,1\}^{T \times N_{HW} \times D_2}$.

**Residual + PWConv.** The attention output is reshaped to the spatial view, processed by PWConv, and fused:

$$X_{\text{fused}} = \text{BN}(\text{PWConv}(\text{Attn})) + X_{\text{in}},$$
$$X_{\text{out}} = \text{MLP}(\text{SN}(X_{\text{fused}})).$$

## F. Experiment Details

### F.1. Training Environment

All experiments were conducted on a single NVIDIA RTX PRO 6000 GPU with 96GB of memory.

### F.2. Surrogate Gradient Function

To address the non-differentiable nature of the spike activation function in direct training, we employed the surrogate gradient method. Specifically, we adopted the sigmoid function as our surrogate function, which is defined as:

$$\text{Sigmoid}(x) = \frac{1}{1 + \exp(-\gamma x)} \tag{41}$$

The steepness parameter $\gamma$ was set to 4.0 in our experiments.

### F.3. Datasets

We employed SpecAugment (Park et al., 2019) and Mixup (Zhang et al., 2017) during training. Detailed experimental settings for each dataset are summarized in Table 6.

**AudioSet.** AudioSet is a large-scale dataset for audio event research, containing 10-second audio clips sourced from YouTube videos. The dataset is organized with an ontology of 527 audio event classes (Gemmeke et al., 2017). It includes a smaller *balanced* subset ( 22k clips) for controlled experiments and an *evaluation* set ( 20k clips). For our work, we utilize this dataset as a foundational resource for training general-purpose audio representations. For our experiments, all audio clips were first resampled to 16 kHz and converted to a single channel (mono). We then extracted log-Mel filterbank energies (Fbank) as input features. The features were computed using a Hanning window with a frame shift of 10 ms. We used 128 Mel frequency bins for the filterbank calculation. To ensure a uniform input size for our model, the resulting feature sequences were padded or truncated to a fixed length of 1024 frames.

**ESC-50 (Environmental Sound Classification).** The ESC-50 dataset is a widely used benchmark for environmental sound classification (Piczak, 2015). It consists of 2,000 5-second audio recordings distributed evenly across 50 distinct semantic classes, such as "Dog" and "Rain," with 40 clips per class. Following standard evaluation protocol, we report the classification accuracy using a 5-fold cross-validation scheme. The audio data from ESC-50 was processed using the same feature extraction methodology applied to AudioSet. After converting the recordings into log-Mel filterbank features, each sequence was padded to a final length of 512 frames.

**Speech Commands V2 (SCV2).** The Speech Commands V2 dataset is designed for keyword spotting tasks (Warden, 2018). It contains approximately 105,000 1-second utterances of 35 short command words. The dataset is pre-divided into standard training, validation, and testing splits. Performance is evaluated based on classification accuracy on the test set. Following the same procedure, we extracted Fbank features from each utterance. The temporal dimension of the resulting feature sequences was then padded to 128 frames to create uniform inputs for our model.

*Table 6.* Experiment details for different datasets.

| Hyperparameter | AudioSet | ESC-50 | SCV2 |
| --- | --- | --- | --- |
| Time step | 4 | 4 | 4 |
| Batch size | 12 | 24 | 96 |
| Optimizer | AdamW | AdamW | AdamW |
| Input shape | (1, 1024, 128) | (1, 512, 128) | (1, 128, 128) |
| **Augmentation** | specAug | specAug + Mixup | specAug + Mixup |
| Epochs | 100 | 200 | 200 |
| Scheduler | Cosine | Cosine | Cosine |
| Warmup epochs | 5 | 5 | 5 |
| Warmup start lr | $1 \times 10^{-3}$ | $1 \times 10^{-4}$ | $1 \times 10^{-4}$ |
| Warmup end lr | $1 \times 10^{-2}$ | $1 \times 10^{-3}$ | $1 \times 10^{-3}$ |
| End lr | $5 \times 10^{-5}$ | $1 \times 10^{-6}$ | $1 \times 10^{-6}$ |

## G. Energy Consumption Estimation Methodology

We estimate the theoretical energy consumption based on the total Synaptic Operations (SOPs) (Kundu et al., 2021; Hu et al., 2021; Yao et al., 2023b; Zhou et al., 2024; 2023), which represent spike-based accumulate (AC) operations, calculated from the FLOPs (floating point operations) of each layer. We distinguish between continuous-valued inputs (MAC operations) and spike-based inputs (AC operations).

The theoretical FLOPs for each layer are calculated according to standard definitions (Molchanov et al., 2017) as follows:

- **Conv2d:** FLOPs $= C_{\text{out}} \times (C_{\text{in}}/g) \times k_h k_w \times H_{\text{out}} W_{\text{out}}$

- **Conv1d:** FLOPs $= C_{\text{out}} \times (C_{\text{in}}/g) \times k \times L_{\text{out}}$

- **Linear:** FLOPs $= \text{in\_features} \times \text{out\_features}$

We adopt the energy constants from a 45nm CMOS process reported in (Horowitz, 2014):

$$E_{\text{MAC}} = 4.6\,\text{pJ} \tag{42}$$

$$E_{\text{AC}} = 0.9\,\text{pJ} \tag{43}$$

1. **SNN Energy:** For spike-driven layers, we scale by time steps $T$ and the layer-wise spike rate $R_\ell \in [0, 1]$, using the energy per accumulate:

$$
\begin{aligned}
\text{SOPs}_\ell &= \text{FLOPs}_\ell \times T \times R_\ell, \\
E_{\text{SNN}} &= \sum_\ell \text{SOPs}_\ell \times E_{\text{AC}}.
\end{aligned}
\tag{44}
$$

We refer to $\text{FLOPs}_\ell \times T \times R_\ell$ as the spiking operation count (SOPs) of layer $\ell$.

2. **Total Energy:** The total energy is obtained by summing the per-layer SNN energies defined above.

## H. Training and Inference Time

The DiceFormer-10-L, DiceFormer-10-M, DiceFormer-10-S, Spikformer, Spike-driven Transformer, Spike-driven Transformer V2 and QKFormer models were trained and evaluated on the AudioSet (20k) dataset. The actual training and inference times, measured on the same device with a batch size (BS) of 12, are detailed in Table 7.

*Table 7.* Training and inference time per epoch for various models on the AudioSet (20k) dataset. All measurements were performed on the same device.

| Model | Training Time (min sec / epoch) | Inference Time (min sec / epoch) |
|---|---|---|
| DiceFormer-10-S | 6m 43s | 2m 32s |
| DiceFormer-10-M | 8m 39s | 3m 23s |
| DiceFormer-10-L | 13m 45s | 5m 16s |
| Spikformer | 9m 33s | 3m 32s |
| Spike-driven Transformer | 9m 15s | 3m 26s |
| Spike-driven Transformer V2 | 12m 58s | 4m 15s |
| QKFormer | 14m 10s | 5m 48s |

## I. Stability Analysis

Table 8 reports the stability of model performance across repeated runs or folds. For AudioSet-20k, we report mAP on a percentage scale over three independent runs. For ESC-50, we follow the official 5-fold protocol and report accuracy averaged across the five folds. For Speech Commands V2 (SCV2), we report accuracy over three independent runs. For all datasets, we provide the mean, standard deviation, and 95% confidence interval.

## J. Correlation Between Attention Score and Spike Density

In this appendix, we analyze the relationship between *attention scores* and *spike density* (firing rate) in spiking Transformer-style audio models trained on AudioSet-20k. We evaluate five representative models: Spikformer, Spike-driven Transformer (SDT v1), Spike-driven Transformer v2 (SDT v2), QKFormer, and DiceFormer. For each model, we compute the Pearson correlation between attention scores and the spike density of the spike tensors involved in computing those scores, and report correlations *per attention stage (layer)*.

Overall, Spikformer and SDT v2 exhibit high positive correlations across stages, SDT v1 and QKFormer show moderate-to-high correlations, and DiceFormer tends to have relatively lower correlations.

*Table 8.* Multi-run stability results across datasets. Performance is reported as mean $\pm$ standard deviation, and the 95% confidence interval (CI) is shown in brackets.

| Dataset | Model | Performance (%) | 95% CI |
|---|---|---|---|
| AudioSet-20k | Spikformer | $13.60 \pm 0.27$ | $[12.93, 14.27]$ |
| AudioSet-20k | Spike-driven Transformer | $13.00 \pm 0.15$ | $[12.63, 13.37]$ |
| AudioSet-20k | Spike-driven Transformer V2 | $12.87 \pm 1.07$ | $[10.21, 15.53]$ |
| AudioSet-20k | QKFormer | $14.70 \pm 0.22$ | $[14.15, 15.25]$ |
| AudioSet-20k | DiceFormer-10-M | $15.42 \pm 0.37$ | $[14.50, 16.34]$ |
| AudioSet-20k | DiceFormer-10-L | $16.11 \pm 0.09$ | $[15.89, 16.33]$ |
| ESC-50 | Spikformer | $45.26 \pm 5.93$ | $[37.89, 52.63]$ |
| ESC-50 | Spike-driven Transformer | $84.79 \pm 3.02$ | $[81.04, 88.54]$ |
| ESC-50 | Spike-driven Transformer V2 | $80.21 \pm 2.85$ | $[76.67, 83.75]$ |
| ESC-50 | QKFormer | $55.26 \pm 14.21$ | $[37.62, 72.90]$ |
| ESC-50 | DiceFormer-10-S | $85.37 \pm 3.05$ | $[81.58, 89.15]$ |
| ESC-50 | DiceFormer-10-M | $85.47 \pm 2.52$ | $[82.33, 88.60]$ |
| ESC-50 | DiceFormer-10-L | $85.73 \pm 2.82$ | $[82.23, 89.23]$ |
| SCV2 | Spikformer | $96.10 \pm 0.28$ | $[95.41, 96.79]$ |
| SCV2 | Spike-driven Transformer | $96.81 \pm 0.17$ | $[96.39, 97.23]$ |
| SCV2 | Spike-driven Transformer V2 | $97.58 \pm 0.36$ | $[96.69, 98.47]$ |
| SCV2 | QKFormer | $97.68 \pm 0.23$ | $[97.11, 98.25]$ |
| SCV2 | DiceFormer-10-S | $97.15 \pm 0.10$ | $[96.91, 97.39]$ |

*Table 9.* Stage-wise Pearson correlation between attention scores and spike density on AudioSet-20k. We report correlations for the first five stages (Stage 0–4) to align with the DiceFormer architecture. "–" indicates that the corresponding stage does not exist for that model configuration.

| Model | Stage 0 | Stage 1 | Stage 2 | Stage 3 | Stage 4 | Mean |
|---|---|---|---|---|---|---|
| DiceFormer-L | 0.1887 | 0.3214 | -0.2203 | 0.1171 | 0.0053 | 0.0824 |
| DiceFormer-M | 0.1552 | 0.3213 | 0.2549 | -0.0496 | 0.3860 | 0.2136 |
| DiceFormer-S | 0.3137 | -0.0658 | -0.3276 | 0.5699 | 0.5080 | 0.1996 |
| QKFormer | 0.6962 | 0.6516 | 0.5699 | – | – | 0.6392 |
| SDT v1 | 0.6705 | 0.6619 | 0.6608 | 0.6490 | 0.6551 | 0.6595 |
| SDT v2 | 0.9287 | 0.9158 | 0.9228 | 0.7955 | 0.9483 | 0.9022 |
| Spikformer | 0.8895 | 0.8398 | 0.8567 | 0.8643 | 0.8557 | 0.8612 |

# K. Neuromorphic Implementation of Spike Dice Attention

## K.1. Spike-driven property of the SDS numerator

### K.1.1. DEFINITION: SPIKE-DRIVEN COMPUTATION.

A computation is considered *spike-driven* if it satisfies the following three conditions:

1. **Binary spike communication:** all neural communication is represented by binary spikes $s \in \{0, 1\}$.

2. **Event-driven processing:** when there is no input spike ($s = 0$), no operation or state update occurs.

3. **Synaptic current form:** the input current $I_i^{\mathrm{cur}}[t]$ can be expressed as a linear combination of spikes $s_j[t]$ and synaptic weights $w_{i,j}$:

$$I_i^{\mathrm{cur}}[t] = \sum_j \left( w_{i,j} \cdot s_j[t] \right). \tag{45}$$

K.1.2. DEFINITION: SPIKE DICE SIMILARITY (SDS).

Let $q, k \in \{0, 1\}^D$ be binary spike vectors of dimension $D$. We define the spike counts

$$
\begin{aligned}
C(q) &:= \|q\|_1 = \sum_{d=1}^{D} q_d, \\
C(k) &:= \|k\|_1 = \sum_{d=1}^{D} k_d,
\end{aligned}
\tag{46}
$$

the overlap

$$
I(q, k) := \|q \odot k\|_1 = \sum_{d=1}^{D} q_d k_d,
\tag{47}
$$

and the total activity $S(q, k) := C(q) + C(k)$. To avoid explicit division, we define the stabilized inverse activity factor

$$
c(q, k) := \big(S(q, k) + \epsilon\big)^{-1},
\tag{48}
$$

where $\epsilon > 0$ is a small constant for numerical stability. Then the SDS score is computed in a multiplication form as

$$
\text{SDS}(q, k) = \big(2I(q, k)\big)\, c(q, k).
\tag{49}
$$

K.1.3. PROPOSITION: SPIKE-DRIVEN PROPERTY OF THE SDS NUMERATOR.

The numerator term $N_{i,j}[t]$ of the SDS score satisfies the spike-driven computation conditions in Section K.1.1. We define the numerator term as

$$
N_{i,j}[t] := 2\, I\big(q_i[t], k_j[t]\big) = 2\sum_{d=1}^{D} \Big(q_{i,d}[t] \odot k_{j,d}[t]\Big).
\tag{50}
$$

K.1.4. PROOF.

We show that the numerator term $N_{i,j}[t]$ satisfies conditions (1)–(3).

**(1) Binary spike communication.** For each channel $d$, define a coincidence signal

$$
s_d[t] := q_{i,d}[t] \odot k_{j,d}[t].
\tag{51}
$$

Since $q_{i,d}[t], k_{j,d}[t] \in \{0, 1\}$, the AND result satisfies $s_d[t] \in \{0, 1\}$. Therefore, the computation uses binary spike signals and satisfies condition (1).

**(2) Event-driven processing.** The coincidence spike $s_d[t]$ is active only when an input event occurs. If there is no input event on channel $d$ (i.e., $s_d[t] = 0$), then the contribution of that channel is zero and no accumulation/state update is triggered. Hence, the numerator accumulation is performed only for channels with coincidence events, satisfying condition (2).

**(3) Synaptic current form.** Using $s_d[t]$, we can rewrite the numerator as

$$
N_{i,j}[t] = 2\sum_{d=1}^{D} s_d[t] = \sum_{d=1}^{D} \big(2 \cdot s_d[t]\big).
\tag{52}
$$

This is mathematically equivalent to the synaptic current form with a fixed weight $w_d = 2$ for each channel:

$$
\sum_{d=1}^{D} \big(w_d \cdot s_d[t]\big) \equiv I_i^{\text{cur}}[t].
\tag{53}
$$

By interpreting the channel index $d$ as the presynaptic index, the coincidence signal $s_d[t]$ as the presynaptic spike, and the constant 2 as the synaptic weight, the numerator term follows the standard spike-driven current accumulation model, satisfying condition (3).

K.1.5. PROPOSITION: SPIKE-DRIVEN PROPERTY OF THE SDS DENOMINATOR (ACTIVITY ACCUMULATION).

We have $q_i[t], k_j[t] \in \{0,1\}^D$. Therefore, the spike-count terms $C\big(q_i[t]\big)$ and $C\big(k_j[t]\big)$ follow the standard spike count–and–accumulate pattern used in existing SNN-Transformers, and can be implemented on neuromorphic hardware in the same manner.

Specifically, by definition,

$$C\big(q_i[t]\big) = \sum_{d=1}^{D} q_{i,d}[t]. \tag{54}$$

Letting the synaptic weight be $w_d = 1$ for each channel, we can rewrite (54) in the synaptic current form as

$$C\big(q_i[t]\big) = \sum_{d=1}^{D} \big(w_d \cdot q_{i,d}[t]\big), \qquad w_d \equiv 1, \tag{55}$$

which matches the spike-driven definition. The term $C\big(k_j[t]\big)$ is derived in an identical spike-driven form:

$$C\big(k_j[t]\big) = \sum_{d=1}^{D} k_{j,d}[t] = \sum_{d=1}^{D} \big(w_d \cdot k_{j,d}[t]\big), \qquad w_d \equiv 1. \tag{56}$$

Consequently, the total activity

$$S\big(q_i[t], k_j[t]\big) := C\big(q_i[t]\big) + C\big(k_j[t]\big) \tag{57}$$

is also obtained by spike-driven accumulations.

## K.2. Implementation of the SDS normalization factor

The SDS normalization factor is defined as

$$c\big(q_i[t], k_j[t]\big) := \big(S\big(q_i[t], k_j[t]\big) + \epsilon\big)^{-1}, \tag{58}$$

which depends only on the scalar activity $S$. Recall that

$$S\big(q_i[t], k_j[t]\big) = C\big(q_i[t]\big) + C\big(k_j[t]\big), \tag{59}$$

and $C(\cdot)$ is obtained by spike-count accumulation. Since $q_i[t], k_j[t] \in \{0,1\}^D$, we have

$$C\big(q_i[t]\big) \in \{0,1,\ldots,D\}, \qquad C\big(k_j[t]\big) \in \{0,1,\ldots,D\}, \tag{60}$$

and therefore

$$S \in \{0,1,\ldots,2D\}. \tag{61}$$

Thus, $c(\cdot)$ can be efficiently realized on neuromorphic hardware using a small lookup table (LUT) indexed by the integer activity $S$.

