# OpenReview forum: "Rethinking Attention in Spiking Transformers: Overcoming Density Bias with Set Similarity"
_ICML.cc/2026/Conference — ICML 2026 regular_

### Official Review · Reviewer_kyBA · 2026-03-09

**Soundness:** 3
**Presentation:** 3
**Significance:** 3
**Originality:** 3
**Overall Recommendation:** 4
**Confidence:** 3

**Summary:**

This paper identifies a limitation of many existing similarity-based spiking attention mechanisms: they can be overly sensitive to spike firing density rather than true alignment. To address this issue, the authors propose DiceFormer, which introduces Dice-based attention formulations such as SDA and Lin-SDA to mitigate density bias in a spike-compatible manner. Experimental results, primarily in the audio domain, show that the proposed approach achieves competitive performance while maintaining promising efficiency.

**Compliance With Llm Reviewing Policy:**

Affirmed.

**Key Questions For Authors:**

1. Most experiments are conducted with a timestep setting of $T=4$. Is there a specific reason for choosing this value? It would also be helpful to understand how the performance and efficiency change as $T$ varies.

**Limitations:**

1. The energy-efficiency claim has not yet been validated on real neuromorphic hardware.

**Strengths And Weaknesses:**

Strengths
1. The paper clearly formulates density bias in spike overlap-based attention as an important problem and proposes SDA/Lin-SDA as an intuitive and well-motivated solution.
2. Beyond introducing a new scoring mechanism, the paper further extends the idea into a hierarchical SADA + SDA architecture that reflects the frequency-temporal structure of audio inputs, which improves the overall design quality.
3. The experimental section is fairly comprehensive, including comparisons on AudioSet-20k, ESC-50, and SCV2, as well as ablation studies, CIFAR-100 transfer experiments, score-density correlation analysis, and latency analysis.

Weaknesses
1. Most of the main experiments are concentrated on audio classification, while evidence beyond the audio domain is limited to a relatively modest drop-in replacement result on CIFAR-100. As a result, the method's generality has not yet been fully established.
2. The efficiency claim is mainly supported by theoretical energy estimation and GPU latency analysis, while validation on actual neuromorphic hardware is left for future work. This limits the practical strength of the hardware-efficiency argument.

---

> ### Author Rebuttal · Authors · 2026-03-31
>
> ### Weakness 1
>
> We agree that the current manuscript does not fully establish strong cross-domain generality of the DiceFormer. However, the main scope of this paper is to extend SNN-Transformers to the audio domain through density-aware spike attention. We therefore hope the CIFAR-100 results can be interpreted not as strong evidence for the full generality of DiceFormer, but as limited yet meaningful evidence that the core SDA can transfer beyond audio. In the final version, we will revise the corresponding wording more carefully so as not to overstate the generality of this evidence.
>
> - **We focused on audio.** Since the main contribution combines an audio-oriented architecture with density-aware spike attention, we focus on representative audio benchmarks with different characteristics: AudioSet-20k, ESC-50, and SCV2.
> - **Meaning of the CIFAR-100 results.** On CIFAR-100, applying SDA as a drop-in replacement to strong SNN vision baselines yields the following consistent gains.
>
> | **Model** | **Accuracy (%)** |
> |---|---|
> | Spikformer [1] | 78.21 |
> | Spike-driven Transformer [2] | 78.40 |
> | Spike-driven Transformer (+SDA) | 78.99 |
> | SNN-ViT [3] | 80.01 |
> | Spikingformer [4] | 80.37 |
> | QKFormer [5] | 81.15 |
> | QKFormer (+SDA) | 81.41 |
>
> - On CIFAR-100, strong SNN baselines already lie within sub-1pp margins, so we view the gains as small but consistent positive shifts under a controlled comparison, and will present them more cautiously as limited evidence that SDA can transfer beyond audio.
> - **Revision in the final version.** We will revise the discussion more carefully so that the main contribution and evidence are clearly framed as audio-domain results, while CIFAR-100 provides only *limited but consistent evidence for SDA transferability*.
>
> ---
>
> ### Weakness 2
>
> We also consider this a valid point. Our efficiency evaluation is based not on physical neuromorphic hardware, but on theoretical energy/SOP analysis and measured GPU runtime, as in prior SNN-Transformer studies [1–5]. Thus, our claim is not superiority on a real chip, but that SDA/Lin-SDA has a hardware-friendly computational structure. Given the limited access to neuromorphic devices and the substantial systems effort required for FPGA-based implementation, such proxy evaluations remain common.
>
> - **Consistency of the evaluation protocol.** Representative prior SNN-Transformer studies have also largely focused on hardware-friendly algorithm design and theoretical energy/SOP-based analysis. Our evaluation is therefore consistent with the protocol commonly used in this line of work.
> - **Why we regard it as hardware-friendly.** The core operations of SDA, namely overlap and total activity, can be computed mainly through bitwise AND, popcount, and addition. In addition, the inverse activity factor can be implemented using a compact LUT based on the bounded activity range, and Lin-SDA approximates this in affine form so that it can be computed without division.
>
> We will revise the hardware-related wording more carefully to make it explicit that the current results are based on proxy/runtime evidence.
>
> ---
>
> ### Question: Why was $T=4$ used?
>
> We used $T=4$ as a practical operating point that enables a fair comparison with prior SNN-Transformer baselines while providing a good balance between performance and efficiency.
>
> - **Meaning of timestep in SNNs.** The timestep $T$ indicates over how many discrete time steps the same input is processed and spikes are accumulated. In general, as $T$ increases, representational power may improve through temporal accumulation, but computational cost and energy consumption also increase.
> - **Why $T=4$ was chosen.** $T=4$ is a representative setting widely used in the existing SNN-Transformer literature [1–5], and we therefore used it as the main setting to preserve comparability with prior baselines.
> - **Trade-off observed in this paper.** For DiceFormer-10-L on AudioSet-20k, we observed 6.18 mJ / 0.153 mAP at $T=1$, and 17.80 mJ / 0.161 mAP at $T=4$. In other words, a smaller $T$ substantially improves efficiency at the cost of lower performance, whereas a larger $T$ provides additional performance at higher cost.
> - **Revision in the final version.** In the final version, we will clarify more explicitly that $T=4$ does not imply a universal optimum, but rather a practical operating point chosen for fair comparison and the performance–efficiency trade-off.
>
> | **Model** | **Timestep ($T$)** | **Energy (mJ)** | **mAP** |
> |---|---|---|---|
> | DiceFormer-10-L | 1 | 6.18 | 0.153 |
> | DiceFormer-10-L | 4 | 17.80 | 0.161 |
>
> ---
>
> **References**
>
> [1] Spikformer: When spiking neural network meets transformer. ICLR, 2023.
>
> [2] Spike-driven Transformer. NeurIPS, 2023.
>
> [3] Spiking Vision Transformer with Saccadic Attention. ICLR, 2025.
>
> [4] Spikingformer: A Key Foundation Model for Spiking Neural Networks. AAAI, 2026.
>
> [5] QKFormer: Hierarchical Spiking Transformer using Q-K Attention. NeurIPS, 2024.

---

### Official Review · Reviewer_64cu · 2026-03-10

**Soundness:** 2
**Presentation:** 3
**Significance:** 2
**Originality:** 3
**Overall Recommendation:** 3
**Confidence:** 4

**Summary:**

This manuscript identifies a critical limitation in existing Spiking Transformer architectures: the density bias in attention mechanisms, where high firing rates spuriously dominate attention scores regardless of semantic relevance. To address this, the authors propose DiceFormer, a hierarchical Spiking Neural Network (SNN) featuring Spike Dice Attention (SDA)—a linear-complexity attention mechanism that employs Dice coefficient-based set similarity to explicitly normalize for spike density. The authors further introduce Spike Audio Dice Attention (SADA) to capture frequency-temporal structures specific to audio spectrograms, and a hardware-efficient variant Lin-SDA. Evaluated on AudioSet-20k, ESC-50, and Speech Commands V2, DiceFormer establishes new state-of-the-art (SOTA) performance among SNNs while narrowing the accuracy gap with Artificial Neural Network (ANN) baselines. The work is supported by theoretical analysis of density bias and comprehensive ablation studies.

**Compliance With Llm Reviewing Policy:**

Affirmed.

**Final Justification:**

I appreciate the additional clarifications provided by the authors. However, their response exhibits a fundamental logical inconsistency: while acknowledging that $C(q)$ and $C(k)$ "may themselves carry task-relevant alignment cues" (implying these counts are potentially structured and non-random), the authors simultaneously condition their null model on fixed values $C(q)=m$ and $C(k)=n$. Statistically, this amounts to treating these information-carrying variables as fixed constants, effectively excluding them from the randomization framework. This selective conditioning creates an illusion of separating "density effects" from "positional effects," when in reality it merely tests for randomness in support placement conditional on fixed (and potentially task-biased) densities—falling far short of a complete null model that would randomize both counts and positions. Furthermore, given that the experimental improvements appear marginal (particularly within the computer vision domain), I will maintain my original score.

**Key Questions For Authors:**

1. Have you evaluated DiceFormer on ImageNet-1K or other large-scale vision benchmarks? If the density bias is a fundamental issue in spiking attention, its impact should be observable in vision domains where SNN Transformers have been most extensively studied. If not, what architectural barriers prevent direct scaling?

2. You mention that the inverse activity factor $c(\cdot)$ can be implemented via a Look-Up Table (LUT) on neuromorphic hardware. Could you specify the hardware platform (e.g., FPGA, ASIC, or specific neuromorphic chip) and provide concrete latency/energy measurements rather than theoretical FLOPs-based estimates? How does the LUT access overhead compare to the savings from avoiding division?

**Limitations:**

yes

**Strengths And Weaknesses:**

**Strengths:**

1. The identification and formal analysis of density bias in spike-based attention represent a meaningful contribution to SNN research.

2. The decoupling of frequency and temporal attention via SADA is well-motivated for audio spectrograms, and the hierarchical design demonstrates careful consideration of domain-specific inductive biases.

**Weaknesses:**

1. Compared to naive spiking self-attention mechanisms, Dice-based Spike Attention introduces an additional inductive bias—specifically, the assumption that attention scores should be independent of spike density (firing rates) in queries and keys. However, this inductive bias is not necessarily correct and may vary across domains, as spike density itself may convey useful information. For instance, self-attention in ANNs employs dot-product operations rather than cosine similarity, where larger vector norms naturally increase dot-product magnitudes, yet this approach has proven highly effective in ViTs and LLMs. Moreover, inductive biases are inherently domain-specific: CNNs leverage local receptive fields, translation invariance, and weight sharing—priors that excel in computer vision yet underperform in NLP where data characteristics violate these assumptions. Additionally, the performance improvements observed on CIFAR-100 (+0.59 and +0.26) are marginal and could readily be attributed to hyperparameter variations (e.g., random seeds, learning rates, or batch sizes) rather than genuine architectural efficacy. Consequently, I remain skeptical of the authors' conclusion regarding the generality of this method across domains.

2. The stability constant $\epsilon$ in the Dice coefficient and the learnable $\lambda$ in Lin-SDA are introduced without analysis of their sensitivity or impact on convergence stability.

---

> ### Author Rebuttal · Authors · 2026-03-31
>
> **Weakness 1.**
> We agree that firing rate can be informative depending on the domain. However, our point is not that attention scores should be fully independent of spike density, but that *density-only inflation* unrelated to alignment should be mitigated.
>
> - **Under the null model in Appendix B.1**, overlap-based spike attention satisfies $\mathbb{E}[I(q,k)] = \frac{C(q)C(k)}{d}$, so even without alignment information, denser keys can be structurally favored.
> - We therefore use $\mathrm{SDS}(q,k)=\frac{2I(q,k)}{C(q)+C(k)+\epsilon}$. Appendix B.2 shows that SDS does not remove density entirely, but yields *diminishing returns* as key density increases, i.e., it mitigates density bias rather than enforcing density independence.
> - Because ANN and SNN attention operate on different input types, overlap-only scores on binary spikes can be biased toward denser queries/keys under null alignment, and SDA is introduced to mitigate this SNN-specific inflation.
> - SDA normalizes only the Q--K score; $V$, the residual path, and the subsequent MLP remain unchanged, so useful density information can still flow through the network.
> - On CIFAR-100, strong SNN baselines already lie within sub-1pp margins, so we view the gains as small but consistent positive shifts under a controlled comparison, and will present them more cautiously as limited evidence that SDA can transfer beyond audio.
>
> | **Model** | **Accuracy (%)** |
> |---|---|
> | Spikformer [1] | 78.21 |
> | Spike-driven Transformer [2] | 78.40 |
> | Spike-driven Transformer (+SDA) | 78.99 (+0.59) |
> | SNN-ViT [3] | 80.01 |
> | Spikingformer [4] | 80.37 |
> | QKFormer [5] | 81.15 |
> | QKFormer (+SDA) | 81.41 (+0.26) |
> ---
>
> **Weakness 2.**
> We agree that the roles of $\epsilon$ and $\lambda$ in Lin-SDA should be clarified, and we will do so in the final version.
>
> - In Eq. (6), $\epsilon=10^{-6}$ is a fixed numerical safeguard to avoid a zero denominator. Since $S(q,k)=C(q)+C(k)\in\{0,\dots,2D\}$ for binary spikes, its effect is negligible except in degenerate cases.
> - $\lambda$ in Lin-SDS is a learnable non-negative parameter from the first-order Taylor approximation in Appendix D: $F(I,S)=\frac{2I}{S}\approx \alpha(I-\lambda S)+\beta$, yielding $\mathrm{SDS}_{\mathrm{Lin}}(q,k)=I(q,k)-\lambda\big(C(q)+C(k)\big)$.
> - **Empirical stability.** The learned layer-wise $\lambda$ stayed in a narrow range of $[0.6929,\,0.6945]$ with no unstable drift or divergence, suggesting stable convergence.
> - This is also consistent with the small gap between Lin-SDA and standard SDA (0.159 vs.\ 0.161 mAP), indicating that the linearized surrogate remains close to the original while behaving stably in optimization.
>
> ---
>
> **Question 1.**
> Thank you for the constructive comment. We did not evaluate the full DiceFormer on ImageNet-1K because the main scope of this paper is audio. Spectrograms have **time** and **frequency** axes with distinct physical meanings unlike natural images, and fine-grained time–frequency structure directly affects key audio characteristics [6,7]. We therefore considered density bias particularly important in audio and designed an audio-centered SNN-Transformer with the SADA module. Table 2 confirms that SDA yields consistent improvements across backbones in the audio setting. Because density bias may not be exclusive to audio, we also evaluated core SDA on vision SNN backbones. We considered CIFAR-100 a reasonable first benchmark to test whether addressing density bias could also improve vision SNNs. Under the same training recipe, Spike-driven Transformer improved from 78.40 to 78.99 and QKFormer from 81.15 to 81.41. We view these as limited but positive evidence that SDA can transfer beyond audio.
>
> **Question 2.**
> We agree that direct hardware validation would be ideal. We followed the common practice in prior SNN studies [1--5] of reporting SOP-based energy estimates together with measured GPU runtime, without targeting a specific FPGA/ASIC/chip platform. We will clarify this limitation more explicitly in the final version.
>
> Regarding LUT overhead, the inverse term depends only on a **bounded activity** value $S(q,k)=C(q)+C(k)\in\{0,\dots,2D\}$, so it can be implemented as a **small reciprocal LUT** indexed by $S$—essentially a single lookup rather than a full divider. We expect LUT overhead to be much smaller than division cost, although the exact trade-off is platform-dependent and remains for future validation.
>
> [1] Spikformer: When spiking neural network meets transformer. ICLR, 2023.
> [2] Spike-driven Transformer. NeurIPS, 2023.
> [3] Spiking Vision Transformer with Saccadic Attention. ICLR, 2025.
> [4] Spikingformer: A Key Foundation Model for Spiking Neural Networks. AAAI, 2026.
> [5] QKFormer: Hierarchical Spiking Transformer using Q-K Attention. NeurIPS, 2024.
> [6] Asit: Local-global audio spectrogram vision transformer for event classification. IEEE/ACM Transactions on Audio.
> [7] MelRe: Vision-Based Mel-Spectrogram Restoration. INTERSPEECH, 2025.

---

> > ### Author Rebuttal · Reviewer_64cu · 2026-04-03
> >
> > Thanks for your responses. A few questions still remain for me:
> >
> > **To W1:**
> > This reasoning contains a logical flaw. Although the mathematical expression $\mathbb{E}[I(q,k)]=\frac{C(q)C(k)}{d}$ holds, the conclusion that "even without alignment information, denser keys can be structurally favored" does not rigorously follow—$C(q)$ and $C(k)$ themselves inherently encode alignment information. For instance, standard self-attention in ANNs also employs dot-product operations, where vector magnitudes $\lVert q\rVert$ and $\lVert k\rVert$ implicitly reflect alignment cues. Furthermore, the notion of "density-only inflation unrelated to alignment" lacks rigorous definition: what are the quantification criteria? How can gains be strictly delineated as arising solely from density? I recommend that the authors provide formal mathematical quantification rather than relying on vague conceptual constructs.
> >
> > **To Q1:**
> > Regarding the experimental design, given that the experimental setup includes CIFAR-100, why was ImageNet not used for validation? CIFAR-100 is notoriously sensitive to hyperparameters, where performance fluctuations from varying random seeds may exceed the gains attributable to the proposed method itself. Consequently, the present evidence is insufficient to substantiate the generalizability of this approach to the visual domain.

---

> > > ### Author Response · Authors · 2026-04-05
> > >
> > > **Response to W1**
> > >
> > > Thank you for this careful and rigorous follow-up. We are grateful to the reviewer for pointing out the lack of mathematical precision in our earlier wording. This helped us refine both the formal modeling and the scope of our claim.
> > >
> > > We agree that our previous phrasing overreached. In particular, the statement that "even without alignment information, denser keys can be structurally favored" does **not** rigorously follow from the expectation formula alone, because $C(q)$ and $C(k)$ may themselves carry task-relevant alignment cues, just as vector magnitudes can in standard ANN dot-product attention. We will therefore remove this wording in the final version.
> > >
> > > Our intended claim is narrower and can be stated more precisely through a count-conditioned null model. Let $q,k \in \\{ 0,1 \\}^d$ denote binary spike vectors, and define their support sets as
> > >
> > > $$S\_q := \\{i \in \\{1,\dots,d\\} \mid q_i = 1 \\}, \qquad S\_k := \\{i \in \\{1,\dots,d\\} \mid k_i = 1 \\}.$$
> > >
> > > $S\_q$ and $S\_k$ are the index sets of active spike locations in $q$ and $k$, respectively. Hence,
> > >
> > > $$|S\_q| = C(q), \qquad |S\_k| = C(k),$$
> > >
> > > and the raw overlap can be written equivalently as
> > >
> > > $$I(q,k)=\sum\_{i=1}^{d} q\_i k\_i = |S\_q \cap S\_k|.$$
> > >
> > > Under a null model in which, conditional on fixed counts $C(q)=m$ and $C(k)=n$, the support locations are sampled uniformly at random from all subsets of $\{1,\dots,d\}$ of sizes $m$ and $n$, respectively, the overlap satisfies
> > >
> > > $$I(q,k)\mid (C(q)=m,\; C(k)=n)\sim \mathrm{Hypergeometric}(d,m,n),$$
> > >
> > > with
> > >
> > > $$\mu(q,k):=\mathbb{E}[I(q,k)\mid C(q)=m,\; C(k)=n] = \frac{mn}{d},$$
> > >
> > > and
> > >
> > > $$\mathrm{Var}[I(q,k)\mid C(q)=m,\; C(k)=n]=\frac{mn(d-m)(d-n)}{d^2(d-1)}.$$
> > >
> > > This formulation does **not** assume that spike counts are unrelated to alignment. Rather, it isolates the overlap expected solely from random support placement **after conditioning on the observed counts**. Thus, the mathematically identifiable quantity is not "density-only inflation" in an absolute sense, but the **count-conditioned baseline**
> > >
> > > $$\mu(q,k)=\frac{C(q)C(k)}{d},$$
> > >
> > > together with the excess overlap above that baseline,
> > >
> > > $$\Delta(q,k)=I(q,k)-\mu(q,k),$$
> > >
> > > or its standardized form
> > >
> > > $$Z(q,k)=\frac{\Delta(q,k)}{\sqrt{\mathrm{Var}[I(q,k)\mid C(q),C(k)]}}.$$
> > >
> > > Accordingly, we will revise our claim as follows: **raw overlap conflates (i) a count-conditioned baseline determined by the marginals and (ii) overlap beyond that baseline.** Under this view, SDS/SDA should be interpreted as reducing direct score-count coupling relative to raw overlap, rather than removing all count effects or proving that the empirical gains arise solely from density.
> > >
> > > We also agree that analogous magnitude coupling can exist in ANN dot-product attention. Our point is therefore not that this issue is unique to SNNs, but that in binary spike space the count-conditioned decomposition naturally motivates count-aware normalization.
> > >
> > > In the final version, we will therefore:
> > > (1) remove wording suggesting "alignment-free inflation,"
> > > (2) present the count-conditioned null model above as the formal quantification criterion, and
> > > (3) revise the discussion so that reduced count-dependent baseline coupling is described as one mathematically identifiable mechanism consistent with the observed gains, rather than the sole explanation.
> > >
> > > We thank the reviewer again for this comment. It helped us clarify the mathematical interpretation and improve the precision of our presentation.
> > >
> > > **Response to Q1**
> > > We thank the reviewer for this important clarification. We agree that ImageNet-1K would be a more appropriate benchmark for a strong vision-domain claim, and that CIFAR-100 alone is insufficient for that purpose, especially given its sensitivity to hyperparameters and random seeds. We acknowledge that our original wording may have overstated the CIFAR-100 results as evidence of generalization beyond audio, and will revise it in the final version.
> > >
> > > Our motivation for including a vision-domain experiment was not to establish robust generalization to vision, nor to claim clear performance gains in that domain. Rather, we intended it only as a preliminary check of whether the core SDA idea can also operate in vision SNN backbones without obvious degradation. The CIFAR-100 experiment was meant as a limited cross-domain check, not as definitive validation of visual-domain generalization.
> > >
> > > Although ImageNet-1K would certainly have been more informative, we prioritized our computational budget toward rigorous validation on the audio datasets, including ablation studies and additional experiments. We will make this limitation explicit in the final version.
> > > In the revised manuscript, we will keep our claims explicitly confined to the audio domain and clarify that the CIFAR-100 experiment was included only to check whether the core SDA idea can be applied to vision SNN backbones without severe performance degradation, rather than to demonstrate generalization in the vision domain.

---

### Official Review · Reviewer_eiu1 · 2026-03-12

**Soundness:** 3
**Presentation:** 3
**Significance:** 3
**Originality:** 3
**Overall Recommendation:** 6
**Confidence:** 4

**Summary:**

This paper proposes DiceFormer, a novel Spiking Transformer architecture designed to address the density bias problem in existing spike-based attention mechanisms. The core contribution is Spike Dice Attention (SDA), which replaces traditional density-sensitive similarity measures with a set-theoretic approach based on the Dice coefficient. By explicitly normalizing for firing density, SDA focuses on spike co-occurrence rather than high firing rates. The paper also introduces Spike Audio Dice Attention (SADA) for capturing frequency-temporal structures in audio data, and Lin-SDA, a hardware-efficient linearized variant. The authors primarily evaluate DiceFormer on audio tasks (AudioSet-20k, ESC-50, Speech Commands V2), where it achieves SOTA performance among SNNs with significantly lower energy consumption compared to ANN baselines. Additionally, the paper demonstrates the generalization of SDA to vision tasks through experiments on CIFAR-100.

**Compliance With Llm Reviewing Policy:**

Affirmed.

**Final Justification:**

Thanks for your rebuttal. My concerns have been solved.

**Key Questions For Authors:**

(1) The density bias analysis assumes uniform distribution of spikes. How robust is the DiceFormer approach when this assumption is violated, such as in cases where spike patterns exhibit strong spatial or temporal correlations? Would the normalization still effectively suppress density bias under highly structured spike distributions?

(2) Please clarify the computational overhead of the SDS normalization. The paper mentions using a LUT for the inverse activity factor, but what is the actual hardware cost of this lookup and the additional accumulation operations compared to standard dot-product attention? How does this scale with feature dimension D?

(3) The Lin-SDA variant shows minimal performance drop (0.002 mAP on AudioSet). Given this small gap, what prevents adopting Lin-SDA as the default? Is there a theoretical or empirical scenario where the full SDS formulation provides substantial benefits over Lin-SDA?

(4) The audio experiments use Mel-spectrogram inputs. Have the authors considered evaluating on raw spike-based audio inputs (e.g., from neuromorphic cochlea sensors) to validate the method's effectiveness in fully event-driven processing pipelines without ANN-based frontend feature extraction?

**Limitations:**

Yes.

**Strengths And Weaknesses:**

Strengths:

(1) The paper shows good originality. It identifies and formally analyzes a fundamental limitation in existing spiking attention mechanisms—density bias—that has been largely overlooked in prior work. The proposed solution using Dice coefficient-based set similarity is both theoretically grounded and practically effective. The extension to audio domain with frequency-temporal decoupling (SADA) represents a meaningful expansion of SNN Transformers beyond vision tasks.

(2) The paper addresses an important problem. Overall, a pressing problem assessed by the paper is that existing spiking attention mechanisms are inherently biased toward high firing rates, causing neurons with dense spike patterns to dominate attention scores regardless of semantic relevance. This is particularly critical for event-driven spiking representations where sparse patterns often carry essential information. This article's important objective is to develop density-aware attention mechanisms that can reliably capture meaningful spike-based relationships while maintaining the energy efficiency benefits of SNNs.

(3) The paper provides rigorous theoretical and empirical analysis. The mathematical derivation of density bias in Appendix B (showing E[I(q,k)] = C(q)C(k)/d) clearly establishes the problem, while the Dice normalization analysis demonstrates how the proposed approach mitigates this bias. The correlation analysis between attention scores and spike density (Figure 2, r = -0.804 with mAP) provides compelling empirical validation. The comprehensive ablation studies and cross-domain validation on CIFAR-100 strengthen the claims.

Weaknesses:

(1) The audio-specific architectural choices lack sufficient ablation. While SADA's frequency-temporal decoupling is intuitively motivated, the paper does not clearly isolate whether the performance gains on audio tasks come primarily from the density-aware SDA mechanism or from the audio-specific architectural design (SADA). A more thorough disentanglement of these factors would strengthen the claims about SDA's general applicability.

(2) The comparison with ANN baselines could be more comprehensive. While the paper shows DiceFormer surpasses scratch-trained AST on AudioSet, the gap with pretrained ANN models (SSAST, DTF-AT) remains substantial. The energy efficiency claims would benefit from more detailed wall-clock time or actual power measurements on neuromorphic hardware, rather than theoretical SOP-based estimates.

(3) There are minor clarity issues in the presentation. The distinction between SDA and SADA could be more explicitly maintained throughout the paper. Some notation in the multi-head formulations (Appendix E) is dense and could benefit from additional explanation. The relationship between the proposed method and prior set-similarity approaches in non-spiking contexts could be discussed more explicitly.

---

> ### Author Rebuttal · Authors · 2026-03-31
>
> **W1. Disentangling the Contributions of SADA and SDA**
>
> Thank you for raising this thoughtful point. Table 2 already provides the key ablation:
>
> | Configuration | mAP | $\Delta$ vs SSA |
> |---|---|---|
> | SDSA only (density-unaware) | 0.126 | -- |
> | SSA only (density-unaware) | 0.142 | -- |
> | SADA only (ours) | 0.157 | +0.015 |
> | SDA only (ours) | 0.158 | +0.016 |
> | SADA $\rightarrow$ SDA, full (ours) | **0.161** | +0.019 |
>
> - **Disentanglement.** SDA only (0.158) performs on par with, or slightly better than, SADA only (0.157), suggesting that density-aware scoring itself is an important contributing factor, rather than the gains coming only from audio-specific decomposition.
> - **Complementarity.** The full model (0.161) outperforms both single-component variants, supporting that SADA and SDA are complementary: SADA provides frequency–temporal inductive bias, while SDA adds density-aware similarity.
> We will clarify this distinction more explicitly in the revision.
>
> ---
>
> **W2. ANN Comparison and Efficiency Claims**
>
> Thank you for this important comment.
>
> - **ANN Comparison.** We agree that a gap to heavily pretrained ANN models remains. However, our goal is to evaluate the proposed attention design in the scratch-trained setting, where its contribution can be interpreted more directly.
> - **Efficiency.** We also agree that neuromorphic-hardware evaluation would be ideal. In practice, such platforms are difficult to access, and full FPGA/neuromorphic implementation is a substantial systems effort. As in many recent SNN studies [1–5], we therefore rely on simulation, SOP-based estimates, and measured GPU runtime. We will clarify this limitation more explicitly.
>
> ---
>
> **W3. Clarity of SDA/SADA and Notation**
>
> We agree and will make the following revisions:
> 1. Consistently distinguish SADA (audio-specific frequency–temporal inductive bias) and SDA (domain-general density-aware scoring) throughout the method description and Figure 1.
> 2. Add a tensor-shape-oriented explanation before the multi-head formulation in Appendix E.
> 3. Clarify SDA's relationship to prior set-similarity measures: the Dice coefficient is classical; our contribution is formulating it as a row-wise density-aware scoring function for spike attention, together with spike-compatible accumulation, compact LUT, and Lin-SDA.
>
> ---
>
> **Q1. Robustness When the Uniform Assumption is Violated**
>
> Thank you for this thoughtful question. The uniform assumption in Appendix B is only a tractable null model for analyzing density bias; it is not required by the method. Its role is to show that higher spike activity can inflate similarity scores. Under structured correlations, the analysis is more involved, but the same first-order density effect can remain, which SDA is designed to normalize. Consistently, DiceFormer performs well on structured real-world datasets (AudioSet-20k, ESC-50, SCV2), where the assumption clearly does not hold. We will clarify this point.
>
> ---
>
> **Q2. Hardware Overhead of SDS Normalization**
>
> The added operations are popcount for $C(\mathbf{q})$, $C(\mathbf{k})$, their sum $S = C(\mathbf{q}) + C(\mathbf{k})$, and a LUT lookup for the inverse factor $c(\cdot)$. Since $S \in [0, 2D]$, the LUT size is $2D{+}1$ (linear in $D$), while the overall complexity remains $O(ND)$. Measured runtime is also consistent with this: DiceFormer (1.67 ms) is comparable to SDT V1 (1.70 ms). Platform-specific LUT cost on neuromorphic devices is planned as future validation.
>
> ---
>
> **Q3. Why Lin-SDA is Not the Default**
>
> Thank you for this thoughtful question. We use full SDA as the default because it is the canonical form of our density-aware similarity and thus the main reference in the same direct-training/GPU setting as prior SNN-Transformers [1–5]. Lin-SDA is a deployment-oriented simplification that preserves most of the gain with lower implementation complexity. Although the gap is small on AudioSet, exact normalization may still be more beneficial under larger activity variation or more heterogeneous sparsity patterns. We will clarify this point.
>
> ---
>
> **Q4. Evaluation on Raw Spike-Based Audio Inputs**
>
> Thank you for this excellent suggestion. We agree that raw spike-based audio would be more appropriate for validating a fully event-driven neuromorphic pipeline. In this paper, we use Mel-spectrograms mainly for fair comparison with prior audio studies and to isolate the effect of the proposed attention mechanism from frontend encoding choices. We agree that end-to-end evaluation with raw event-based audio is an important next step and will clarify this more explicitly.
>
> ---
>
> **References**
>
> [1] Spikformer: When spiking neural network meets transformer. *ICLR*, 2023.
>
> [2] Spike-driven Transformer. *NeurIPS*, 2023.
>
> [3] Spiking Vision Transformer with Saccadic Attention. *ICLR*, 2025.
>
> [4] Spikingformer: A Key Foundation Model for Spiking Neural Networks. *AAAI*, 2026.
>
> [5] QKFormer: Hierarchical Spiking Transformer using Q-K Attention. *NeurIPS*, 2024.

---

> > ### Author Rebuttal · Reviewer_eiu1 · 2026-04-03
> >
> > Thanks for the rebuttal

---

> > > ### Author Response · Authors · 2026-04-03
> > >
> > > Thank you for the careful follow-up and for taking the time to revisit the paper. We sincerely appreciate your positive reassessment. We will incorporate these clarifications and additions into the revised version of the paper.

---

### Official Review · Reviewer_rs7M · 2026-03-19

**Soundness:** 3
**Presentation:** 3
**Significance:** 2
**Originality:** 3
**Overall Recommendation:** 4
**Confidence:** 2

**Summary:**

This paper introduces DiceFormer, a spiking neural network (SNN) transformer with Spike Dice Attention (SDA) - a mechanism that reduces the sensitivity of attention scores to high spike rates in existing spiking transformers. A computation-efficient Lin-SDA is also introduced. DiceFormer shows state-of-the-art performance on AudioSet-20k, approaching artificial neural networks (ANN) performance while consuming less energy.

**Compliance With Llm Reviewing Policy:**

Affirmed.

**Final Justification:**

The authors made efforts to adequately address my concerns on the comprehensiveness of the performance benchmarking. I've increased my scores.

**Key Questions For Authors:**

Could the authors show the performance of all baselines across all datasets for comprehensive comparison?

**Limitations:**

The authors did not adequately discuss the limitations and potential negative societal impact of their work.

**Strengths And Weaknesses:**

Strengths:
- DiceFormer is well motivated. It addresses a weakness in spiking transformers where high spike rates dominate attention scores regardless of semantic relevance, which it demonstrates to be harmful to model performance.
- The experimental results largely support the claims with comparison against multiple baselines.
- The figures and writing are well organized and easy to follow.

Weaknesses:
- The set of DiceFormer variants and baselines evaluated in each of the audio datasets (AudioSet-50k, ESC-50, SCV2) are inconsistent, making it hard to know if the performance gain over each baseline is consistently observed across datasets.
- There is no confidence interval or statistical test on the reported performance, making it hard to judge the significance of the performance improvement over baselines.
- The improvement in accuracy on CIFAR-100 with drop-in SDA seems to be negligible (Table 3).

---

> ### Author Rebuttal · Authors · 2026-03-31
>
> **Weakness 1. Inconsistent baseline coverage across datasets**
>
> Thank you for this helpful comment. AudioSet-20k included the main SNN Transformer baselines, whereas ESC-50 and SCV2 only included a subset. On smaller benchmarks, we kept each model's original backbone unchanged under the same from-scratch setting, rather than introducing dataset-specific modifications to vision-first SNN Transformers. ESC-50 is extremely small (40 samples per class), and under the same from-scratch recipe some large vision-first SNN baselines showed substantial underfitting, making comparison harder to interpret. Prior SNN literature also often uses task-dependent configurations on smaller-scale benchmarks [1–5]. That said, we fully agree that denser comparison is important, and we performed additional experiments within the rebuttal period, using representative SNN baselines.
>
> - **Additional experiments.** We evaluated **DiceFormer-10-L** on ESC-50, and **Spikformer and QKFormer** on SCV2.
> - **Results.** DiceFormer still shows the strongest performance on ESC-50. On SCV2, our previously reported DiceFormer-10-S (97.27%, 13.5M) remains competitive with the large baselines while using substantially fewer parameters.
> - **Revision.** We will include a unified cross-dataset comparison table and clarify the rationale and limitation of the original baseline coverage.
>
> | **Dataset** | **Model** | **Score (%)** | **Params (M)** |
> |---|---|---|---|
> | ESC-50 | DiceFormer-10-L | 85.73 | 59.3 |
> | SCV2 | Spikformer | 96.10 | 65.6 |
> | SCV2 | QKFormer | 97.68 | 64.2 |
>
> ---
>
> **Weakness 2. Lack of confidence intervals or statistical tests**
>
> Thank you for this helpful comment. Direct statistical testing against baselines would require repeated runs for each comparison model, while most prior ANN [6–8] and SNN [1–5] baselines do not report such statistics. Retraining all baselines multiple times within the rebuttal period was not feasible. Nevertheless, we conducted additional variability analysis for our own model.
>
> - **ESC-50 (5-fold):** DiceFormer-10-S: $85.37 \pm 3.05$, 95% CI [81.58, 89.15]; DiceFormer-10-M: $85.47 \pm 2.52$, 95% CI [82.33, 88.65]; DiceFormer-10-L: $85.73 \pm 2.82$, 95% CI [82.23, 89.23]
> - **SCV2 (3-run):** DiceFormer-10-S: $97.15 \pm 0.10$, 95% CI [96.91, 97.39]
> - **Key observation.** Mean performance remains close to the originally reported scores, supporting that DiceFormer is generally stable rather than dependent on a single favorable run.
> - **Revision.** We will include these statistics and clarify that direct significance tests against baselines were not feasible within the rebuttal period.
>
> ---
>
> **W3. Magnitude of CIFAR-100 improvements**
>
> Thank you for this helpful comment. We agree the improvements on CIFAR-100 are not large. However, replacing only the attention module with SDA under the same backbone and training recipe yields consistent gains on two strong SNN baselines.
>
> | **Model** | **Accuracy (%)** |
> |---|---|
> | Spikformer [1] | 78.21 |
> | Spike-driven Transformer [2] | 78.40 |
> | Spike-driven Transformer (+SDA) | 78.99 (+0.59) |
> | SNN-ViT [3] | 80.01 |
> | Spikingformer [4] | 80.37 |
> | QKFormer [5] | 81.15 |
> | QKFormer (+SDA) | 81.41 (+0.26) |
>
> - **Key point.** Since strong SNN baselines on CIFAR-100 already differ within sub-1pp margins, we view these as small but consistent positive shifts under a controlled comparison.
> - **Revision.** We will present the CIFAR-100 result more cautiously as limited supportive evidence for the cross-domain applicability of SDA.
>
> ---
>
> **W4. Limitations / societal impact**
>
> Thank you for this important comment. We agree that limitations and societal impact should be stated more explicitly in the final version.
> - **Revision.** Specifically, we will clarify that (i) our hardware-efficiency discussion is based on a theoretical energy proxy and measured GPU runtime, while direct validation on actual neuromorphic hardware remains future work, and (ii) like other audio classification models, our model can still make incorrect predictions and should not be used for high-stakes decision-making without human oversight.
>
> **References**
>
> [1] Spikformer: When spiking neural network meets transformer. ICLR, 2023.
>
> [2] Spike-driven Transformer. NeurIPS, 2023.
>
> [3] Spiking Vision Transformer with Saccadic Attention. ICLR, 2025.
>
> [4] Spikingformer: A Key Foundation Model for Spiking Neural Networks. AAAI, 2026.
>
> [5] QKFormer: Hierarchical Spiking Transformer using Q-K Attention. NeurIPS, 2024.
>
> [6] AST: Audio Spectrogram Transformer. INTERSPEECH, 2021.
>
> [7] SSAST: Self-supervised audio spectrogram transformer. AAAI, 2022.
>
> [8] DTF-AT: Decoupled Time-Frequency Audio Transformer for Event Classification. AAAI, 2024.

---

> > ### Author Rebuttal · Reviewer_rs7M · 2026-04-04
> >
> > I thank the authors for their rebuttal. My concerns over the comprehensiveness of model benchmarking against all baselines, however, remain unresolved. The paper might benefit from another round of revision to get complete results on baseline performance and confidence intervals.

---

> > > ### Author Response · Authors · 2026-04-07
> > >
> > > Thank you very much for the helpful follow-up.
> > >
> > > Following the reviewer’s suggestion, we have now completed the previously missing evaluations and provide a consolidated comparison against representative baselines across datasets. The updated table reports the corresponding mean/std and 95% confidence intervals. AudioSet-20K and SCV2 results are averaged over three independent runs, while ESC-50 follows the official 5-fold cross-validation protocol.
> > >
> > > **Table: Additional cross-dataset evaluations and repeated-evaluation statistics added during the rebuttal period. Performance is measured by mean average precision (mAP) for AudioSet-20K and classification accuracy for ESC-50 and SCV2.**
> > >
> > > | **Dataset** | **Model** | **Performance (%)** | **95% CI** |
> > > |---|---|---|---|
> > > | AudioSet-20K | Spikformer | $13.60 \pm 0.27$ | [12.93, 14.27] |
> > > | AudioSet-20K | Spike-driven Transformer | $13.00 \pm 0.15$ | [12.63, 13.37] |
> > > | AudioSet-20K | Spike-driven Transformer V2 | $12.87 \pm 1.07$ | [10.21, 15.53] |
> > > | AudioSet-20K | QKFormer | $14.70 \pm 0.22$ | [14.15, 15.25] |
> > > | AudioSet-20K | DiceFormer-10-M | $15.42 \pm 0.37$ | [14.50, 16.34] |
> > > | AudioSet-20K | DiceFormer-10-L | $16.11 \pm 0.09$ | [15.89, 16.33] |
> > > | ──────── | ──────────────────── | ──────────── | ──────── |
> > > | ESC-50 | Spikformer | $45.26 \pm 5.93$ | [37.89, 52.63] |
> > > | ESC-50 | Spike-driven Transformer | $84.79 \pm 3.02$ | [81.04, 88.54] |
> > > | ESC-50 | Spike-driven Transformer V2 | $80.21 \pm 2.85$ | [76.67, 83.75] |
> > > | ESC-50 | QKFormer | $55.26 \pm 14.21$ | [37.62, 72.90] |
> > > | ESC-50 | DiceFormer-10-S | $85.37 \pm 3.05$ | [81.58, 89.15] |
> > > | ESC-50 | DiceFormer-10-M | $85.47 \pm 2.52$ | [82.33, 88.60] |
> > > | ESC-50 | DiceFormer-10-L | $85.73 \pm 2.82$ | [82.23, 89.23] |
> > > | ──────── | ──────────────────── | ──────────── | ──────── |
> > > | SCV2 | Spikformer | $96.10 \pm 0.28$ | [95.41, 96.79] |
> > > | SCV2 | Spike-driven Transformer | $96.81 \pm 0.17$ | [96.39, 97.23] |
> > > | SCV2 | Spike-driven Transformer V2 | $97.58 \pm 0.36$ | [96.69, 98.47] |
> > > | SCV2 | QKFormer | $97.68 \pm 0.23$ | [97.11, 98.25] |
> > > | SCV2 | DiceFormer-10-S | $97.15 \pm 0.10$ | [96.91, 97.39] |
> > >
> > > These added results directly address the reviewer’s concern. The overall conclusion is further strengthened: DiceFormer shows competitive performance relative to representative baselines across datasets. Furthermore, its advantage on the primary benchmark (AudioSet-20K) is now supported by repeated evaluation and confidence intervals. We sincerely appreciate this suggestion, as it helped us significantly improve the completeness and reliability of the empirical evaluation. We will incorporate these additions and clarifications into the revised manuscript.

---

### Decision · Program_Chairs · 2026-04-30

**Decision:**

Accept (regular)

**Comment:**

This paper identifies and analyzes the "density bias" problem in existing Spiking Transformers attention mechanisms and proposes a set similarity attention mechanism (SDA) based on Dice coefficients to mitigate it, achieving competitive performance on audio benchmarks such as AudioSet-20k. During the peer review process, most reviewers acknowledged the originality of this research perspective and appreciated the detailed cross-dataset baseline comparisons provided by the authors during the rebuttal stage. Although one reviewer fundamentally raised well-founded theoretical questions about the "density bias elimination" assumption, arguing that the impulse density itself may carry important alignment information and pointing out the limitations of the null model derivation, considering the authors' objective acknowledgment of the limitations of this method's generalization ability in the visual domain and the purely theoretical nature of its energy efficiency evaluation, overall, this paper makes a solid exploratory contribution to the SNN community and is therefore recommended for Weak Acceptance.